ecology/evolution/health and disease and epidemiology

host–parasite relationships, parasite survival, propagule persistence, transmission, trade-off theory, virulence evolution

**Author for correspondence:**
Wendy C. Turner
e-mail: wendy.turner@wisc.edu

# The roles of environmental variation and parasite survival in virulence–transmission relationships

Wendy C. Turner[1], Pauline L. Kamath[3], Henriette van Heerden[4], Yen-Hua Huang[2], Zoe R. Barandongo[2], Spencer A. Bruce[5] and Kyrre Kausrud[6]

[1]US Geological Survey, Wisconsin Cooperative Wildlife Research Unit, Department of Forest and Wildlife Ecology, and [2]Department of Forest and Wildlife Ecology, University of Wisconsin-Madison, Madison, WI 53706, USA
[3]School of Food and Agriculture, University of Maine, Orono, ME 04469, USA
[4]Faculty of Veterinary Science, Department of Veterinary Tropical Diseases, University of Pretoria, Onderstepoort, South Africa
[5]Department of Biological Sciences, University at Albany, State University of New York, Albany, NY 12222, USA
[6]Section for Epidemiology, Norwegian Veterinary Institute, Ullevålsveien 68, 0454 Oslo, Norway

WCT, 0000-0002-0302-1646; PLK, 0000-0002-6458-4514;
HvH, 0000-0002-3577-1273; Y-HH, 0000-0002-2961-0895;
ZRB, 0000-0001-6332-991X; SAB, 0000-0002-9423-8049;
KK, 0000-0002-2738-7351

Disease outbreaks are a consequence of interactions among the three components of a host–parasite system: the infectious agent, the host and the environment. While virulence and transmission are widely investigated, most studies of parasite life-history trade-offs are conducted with theoretical models or tractable experimental systems where transmission is standardized and the environment controlled. Yet, biotic and abiotic environmental factors can strongly affect disease dynamics, and ultimately, host–parasite coevolution. Here, we review research on how environmental context alters virulence–transmission relationships, focusing on the off-host portion of the parasite life cycle, and how variation in parasite survival affects the evolution of virulence and transmission. We review three inter-related 'approaches' that have dominated the study of the evolution of virulence and transmission for different host–parasite systems: (i) evolutionary trade-off theory, (ii) parasite local adaptation and (iii) parasite phylodynamics. These approaches consider the role of the environment in virulence and transmission evolution from different angles, which entail different advantages and potential biases. We suggest improvements to how to investigate

virulence–transmission relationships, through conceptual and methodological developments and taking environmental context into consideration. By combining developments in life-history evolution, phylogenetics, adaptive dynamics and comparative genomics, we can improve our understanding of virulence–transmission relationships across a diversity of host–parasite systems that have eluded experimental study of parasite life history.

# 1. Introduction: parasites, hosts and their environment

The 'epidemiological triangle' recognizes that disease outbreaks depend on all three components of a host–parasite system: the infectious agent, the host and the environment. As a result, effective disease management hinges on our ability to estimate key disease parameters (e.g. transmission rate, $R_0$, infection duration, force of infection (see box 1 for definitions)), and to understand the mechanisms creating variation in these parameters [2]. To quote Ovaskainen & Laine [3]: 'one of the principal challenges in epidemiological analysis is to understand the causes of the variability that occurs between different epidemics of the same disease'. This review, therefore, has two main goals. The first is to draw attention to the importance of the off-host portions of parasite life cycles in causing variation in host–parasite relationships and disease outbreaks. To achieve this, we broadly review how environmental context alters host–parasite dynamics, and hone in on under-studied parasite off-host survival traits and their ecological and evolutionary implications for virulence–transmission relationships. The second goal is to suggest improvements for how we investigate virulence–transmission relationships, by taking the environmental context of parasite transmission into consideration and including a greater diversity of host–parasite taxa.

Evolutionary biologists have long been interested in life-history trade-offs in host–parasite systems and their implications for coevolution and outbreak dynamics (throughout this review, we use the term 'parasite' in the general sense, to include microparasites and macroparasites). The trait most commonly investigated is virulence—generally seen as the parasite-induced host mortality or morbidity (see box 2 for variable definitions and their implications)—due to the public health and veterinary implications of host health. However, transmission is a similarly complex process (see box 3), subject to heterogeneities in host, parasite and the external environment, which can create nonlinearities in the transmission process [11]. Recent insights into parasite evolution emphasize the importance of considering both within-host infection dynamics and between-host transmission dynamics for predicting virulence evolution and epidemiology [9,17–19]. In other words, we may only recently have understood the complexity of the challenges inherent in detecting and predicting life-history trade-offs for a diversity of host–parasite systems.

We know from emerging infectious diseases that host–parasite interactions can evolve rapidly, leading to unforeseen changes in disease severity [20–24]. As an example, *Bacillus cereus*, a close relative of *B. anthracis*, has repeatedly acquired plasmids that allow the expression of an anthrax-like phenotype [25], showing how highly pathogenic disease agents can develop where the environment is conducive to such life-history strategies [26–29]. This is not only evident for recently emerged infections in naive host populations, but can be detected in coevolutionary systems with a long history, such as the plague bacterium, *Yersinia pestis* [30–34]. New research suggests *Y. pestis* repeatedly evolved novel traits to enhance its transmission from arthropod vector to mammalian host under cold and dry climate conditions when vector populations were depressed [35]—highlighting how environmental conditions can lead to evolution in transmissibility traits.

There are many sources of variation that can affect infection parameters, including heterogeneities among individual hosts [36,37], parasite strains [38,39] and environments [40]. Environmental factors, including both abiotic and biotic factors, can lead to context-dependent parasitism and virulence [41–43]. Abiotic factors clearly affect both host and parasite life cycles, and thus outbreak epidemiology [44–48], so climate variables are often used to predict parasite distributions [49,50] and forecast disease outbreaks [51,52]. Biotic factors also affect host–parasite interactions, including host variation and a variety of species interactions occurring inside and outside of hosts (e.g. [53–57]). It is important to remember that the environment contributes to phenotype, and that phenotypic plasticity is modulated through epigenetic variation [58–60] meaning that the effects of environmental variation are not necessarily static.

In the first half of this review, we highlight three main research 'approaches' that have dominated the study of the evolution of virulence and transmission for different host–parasite systems. These approaches treat the environment very differently when considering evolutionary interactions between hosts and parasites, ranging from ignoring or controlling for the environment, to using the

**Box 1.** Glossary.

*Environmental transmission:* transmission where hosts are infected via contact with infectious propagules in an environmental reservoir, rather than from contact with another host or vector.

*Facultative, or opportunistic, parasites:* These organisms do not require host exploitation in order to reproduce, but can parasitize organisms as opportunity presents (see Brown *et al.* [1]). Examples include environmental opportunists such as *Flavobacterium columnare*, *Listeria monocytogenes* and *Clostridium* spp. and commensal opportunists such as *Staphylococcus aureus*, *Enterococcus faecalis* and *Mycoplasma ovipneumoniae*.

*Force of infection:* the rate at which susceptible individuals acquire an infection.

*Genotype by environment interactions:* When different genotypes react to changes in the environment in different ways. For example, one genotype may have higher fitness than another genotype under one environmental condition, but those relationships may be reversed under different conditions.

*Indirect transmission:* host infection via contact with an environmental reservoir, an arthropod vector or an intermediate host.

*Infectivity:* a parasite's ability to establish an infection. In plant pathology, this is the proportion of host genotypes a parasite is capable of infecting, often termed virulence (see box 4).

*Life-history traits:* Phenotypes that contribute to the fitness (evolutionary success) of an organism. These include traits that characterize the timing of growth, reproduction, survival. For parasites, the key traits examined are reproduction in the host (in many studies conflated with, or used as a proxy for, virulence) and transmission from one host to the next (see boxes 2 and 3).

*Obligate killer parasites:* These obligate parasites must kill the host to transmit. Examples of this burst transmission include insect viruses such as baculovirus or nucleopolyhedrovirus, bacteria such as *Pasteuria ramosa* and *Bacillus anthracis*, protists such as *Paranosema whitei*, and fungi such as *Metarhizium anisopliae*.

*Obligate shedding parasites:* a parasitic organism that cannot complete its life cycle without exploiting a suitable host that sheds reproductive or infectious stages during the course of the infection. Examples include many respiratory infections spread through coughing and sneezing, as well as gastrointestinal macroparasites and coccidia that require a developmental period in the environment before becoming infectious.

*Parasite:* here broadly defined to include both microparasites (e.g. viruses, bacteria, fungi and protists) and macroparasites (e.g. helminths, arthropods).

*Pathogenicity:* the harm a parasite causes to the host, a term often used interchangeably with virulence (see box 2).

*Phylodynamics:* the study of how interactions between parasite epidemiological and evolutionary processes shape parasite phylogenies, which is often applied as a framework for examining transmission dynamics.

*Phylogeography:* a field combining phylogenetics with biogeography, to investigate relationships in the distribution of genetic lineages on spatial scales.

$R_0$: the basic reproduction number of an infection; an estimate of the average number of secondary cases caused by a single case in a wholly naive population. This gives an estimate of how quickly a disease will spread if a time frame per infection is implied (see box 3).

*Transmission potential:* A term with mixed and somewhat imprecise definitions, representing the likelihood of future transmission events. This is applied both for interactions between different species in vector-borne and zoonotic diseases (encompassing traits such as vector competence and host susceptibility), and as an assessment of parasite fecundity (i.e. the number of propagules produced during an infection, with the arguably simplistic assumption that more propagules produced equates to more future cases, see box 3).

*Virulence:* an emergent property of the host–parasite interaction; the negative effect of the parasite on host fitness (see box 2 for additional discussion).

environment as an experimental treatment for host–parasite coevolution. The approaches include (i) theoretical development and experimental testing of *evolutionary trade-off theory* for virulence and transmission traits, (ii) empirical studies of parasite *local adaptation* across space, and (iii) genetic studies of parasite *phylodynamics*, contrasting transmission dynamics through time. We categorize

> **Box 2.** Challenges of defining and measuring virulence.
>
> One of the great challenges for generalizing virulence–transmission relationships across a range of host–parasite taxa is the variable definitions used to define and measure traits for virulence (and transmission, see box 3), and the scales at which these are applied (populations versus individuals).
>
> Virulence is an emergent property of the host–parasite–environment interaction [4], with broad and often conflicting definitions. In general, parasite virulence encompasses the processes of infectivity, replication within the host, and damage caused to the host, and at evolutionary scales, the fitness of the parasite and its host [5]. Virulence is often treated as a parasite property, yet indirectly measured through its effects on host fitness, i.e. parasite pathogenicity or damage to the host [6–8].
>
> Different fields and approaches vary in which part of this virulence process is emphasized and measured. In theoretical studies, virulence usually refers to the fitness cost a pathogen imposes on a host population and is described as an increase in host mortality rate [7,9]. However, this definition—the most commonly used concept of virulence—combines and conflates the effects of host tolerance/resistance and parasite phenotype [10], arguably with detrimental effects to our understanding of parasite evolution, such as the potential trade-offs in virulence and transmission. For empirical systems, we are limited by what can be readily measured. In experimental systems, virulence is often measured as the parasite-induced host mortality rate. Outside of controlled laboratory systems, proxies for virulence that are commonly measured include parasite growth rate, time to host death, number of parasite propagules produced from an infection, host morbidity (a parasite-induced reduction in host fitness) and intra-infection dynamics (competition or cooperation depending on parasite relatedness).

these overlapping viewpoints into 'approaches', not to define methodologies, but as a useful lens through which to compare these perspectives. They draw upon the same body of theory, but diverge based on the taxa most commonly studied, as well as conventions, vocabularies and methodologies. The focus on different study systems and tool kits has shaped the lenses through which these problems have been viewed, resulting in somewhat different approaches to fundamentally the same questions. We describe how these approaches consider the role of the environment in virulence and transmission evolution from different angles, which entail different advantages and potential biases.

In the second half of this review, we offer some perspectives on how to integrate these approaches to investigate virulence–transmission relationships across the range of biodiversity in host–parasite systems. This section focuses broadly on the pivotal role of the environmental context in disease processes, and specifically on parasite dynamics in the off-host environment. We note the continued need for more interdisciplinary research, as well as conceptual and methodological developments in combination with advances in life-history evolution, phylogenetics, adaptive dynamics and comparative genomics, to improve our understanding of virulence–transmission relationships.

# 2. Approach I: theories linking parasite virulence and transmission (and their mixed empirical support)

The first approach to the study of the evolution of virulence and transmission in host–parasite systems combines the theoretical foundation of life-history relationships between virulence and transmission traits in host–parasite systems, with empirical tests of these theories. Virulence and transmission traits are important to consider when managing infectious diseases, especially given the risk for unintended consequences of interventions on disease outcomes [61–63]. As evidence of the collective interest in virulence and transmission relationships, a recent meta-analysis reviewed over 6000 publications on virulence life-history trade-offs [64]. Under the prevailing theory, parasites are expected to evolve intermediate levels of virulence due to a trade-off between the parasite's virulence and its transmissibility [65,66]. Causing too much harm to the host runs the risk of killing one host before being transmitted to the next, or ultimately driving the host to extinction. Conversely, causing too benign an infection is expected to decrease the transmission rate, creating a competitive disadvantage in comparison with more virulent strains.

These expectations, although framed generally, are focused on directly transmitted parasites, where infectious stages are shed during the infection, and thus virulence and transmission are directly

**Box 3.** Challenges of defining and measuring transmission (and its relationship to virulence).

Like virulence, transmission is a combination of host and parasite traits, including host susceptibility, parasite infectivity, transmission routes, contact rates and exposure doses [11,12], all of which may be under selection. This includes everything from the number of propagules produced during the infection, to the mode and timing of their release into the environment, their persistence in the off-host environment, contact with a susceptible host, dose encountered, parasite infectiousness and heterogeneity in host resistance. Given the challenges of measuring transmission and its heterogeneities, this process is often subject to simplifying assumptions, several of which we discuss here.

Estimates of $R_0$ are often used as a proxy measure of parasite fitness, yet care must be taken as this is not the same as $R_0$ in the epidemiological sense. $R_0$ is fundamentally unsuitable as a fitness measure as long as it ignores the time axis between infections: parasite strain $a$ may start fewer new infections on average than parasite strain $b$, but may still outcompete $b$ if hosts are infected over a shorter time period, $T$. This is because if the number of hosts, $H$, infected by a strain in time $T$ is $H_a = R_{0,a}^{T/Ta}$, and $H_b = R_{0,b}^{T/Tb}$, then $a$ produces more infections as long as $\ln(R_{0,a}) > (T_a/T_b)\ln(R_{0,b})$ when $R_0$ does not include time explicitly.

A second assumption is that the number of propagules released during an infection linearly correlates with transmission success, an assumption that is problematic in several ways:

— A higher quantity of propagules does not necessarily enhance transmission success if there is a trade-off between propagule quality and quantity. For example, environmental survival of *Escherichia coli* bacteriophages decreases with the number of virions produced [13].

— Propagules are not released into a well-mixed pool with equal opportunity to infect a susceptible host. Transmission will be affected by the spatial properties of propagule release from infected hosts. Propagules shed during infections will release relatively smaller numbers of infectious propagules over a larger area, while obligate killer infections release relatively larger concentrations of propagules in a small area.

— Environmental heterogeneity will affect propagule survival in the off-host environment, where a multitude of abiotic and biotic factors can alter parasite development and survival, and hence transmission. Environmental conditions affect transmission processes at a variety of spatial scales, including the micro-scale effect on propagule persistence, the landscape-scale effect on host density and behaviour, and broad-scale biogeographic patterns in host and parasite range. Many studies have highlighted heterogeneity in parasite risk among individual hosts (e.g. [14–16]), but few, in turn, examine where propagules are shed, and how their distribution affects parasite survival and development, host contact and parasite selection.

In the literature review conducted for this study (summarized in table 1), 25% of the studies on virulence/transmission evolution considered for inclusion used the number of propagules produced during an infection as both a virulence trait and, implicitly, a transmission trait (i.e. the 'transmission potential'). Using the same measure to represent virulence and transmission is potentially problematic when determining evolutionary relationships between the two processes.

Assuming that more propagules released equals higher rates of transmission is a simplistic representation of a complex process, especially for environmentally transmitted parasites (ETPs) that can persist for extended periods in the environment and whose success depends on the spatio-temporal distributions of parasites, hosts and contact rates.

coupled through host morbidity and mortality. Yet, transmission mode has been repeatedly linked with virulence, where parasites with indirect transmission, in general, are more virulent than directly transmitted parasites [67–71]. Throughout this review, we use the term environmental transmission to describe host infection from an environmental reservoir (others would call this indirect transmission, but this term has a broader definition, e.g. [72]). Parasites with mixed transmission modes also show heightened virulence when maintaining an environmental transmission pathway (e.g. *Vibrio cholerae* [73], avian influenza viruses [74], *Flavobacterium columnare* [75]).

Environmentally transmitted parasites (ETPs), on the other hand, have been assumed to exhibit different life-history relationships between virulence and transmission. For these parasites, the transmission component most often considered is parasite survival in the environment, as a distinctive

**Box 4.** Challenges across the host taxonomic gulf.

Among host–parasite interactions, a variable coevolutionary landscape altering species interactions across spatial scales has been best recognized for plant pathogens [108,109] and macroparasites, especially of invertebrate hosts [110–112]. Like the trade-off literature described under Approach I, the plant pathology literature also tends to measure parasite fitness traits during the within-host phase (host infection), and not the between-host stage (transmission). In plant pathology, the specific parasite fitness traits measured include infectivity and aggressiveness [113]. Infectivity is the proportion of host genotypes a parasite is capable of infecting, and is often termed virulence. Aggressiveness is further parasite growth and development that determine its 'transmission potential' (i.e. number of propagules produced). Under these definitions, infectivity is considered a virulence trait, although it can also be considered part of the transmission process, and aggressiveness is more similar to measures of virulence for animal microparasites. These differences in definitions are one of several barriers between plant and vertebrate disease researchers. Although basic principles of parasite–host interactions should apply to any host system, there are important differences between vertebrates and plants that have led to a divergence in disease ecology research depending on host taxa, such as the differences in host mobility, response to infection and host trophic level (producers versus consumers) [114], differences which alter research methods and basic predictions about transmission dynamics, coevolution and community-wide effects of parasitism.

Work on mammal diseases, especially microparasitic diseases, have on the other hand been slow to embrace an evolutionary approach to disease transmission dynamics [115], such as those examined under Approach II. There have been few parasite local adaptation studies done on vertebrate hosts (e.g. [116]), in part due to their longer generation times and greater mobility, exposing hosts to a greater range of environmental conditions than experienced by shorter-lived, less mobile hosts. This increased complexity would make detecting genotype by environment interactions increasingly difficult in these organisms, without studies covering larger temporal and spatial scales. The experimental methodology used to test local adaptation comprises reciprocal transplant or common garden experiments, but these are generally impossible to conduct on large, mobile wildlife. Exploring how the environment shapes virulence–transmission relationships for real systems with large mobile hosts is made even more difficult by these hosts being further subject to environmental variation shaping their grouping and movement behaviour [117,118]—both relevant to disease transmission. A variable environment will also alter host immune response to infection [119–121].

trait of these parasites and one that is relatively easy to quantify in experimental studies. Hypotheses for ETP virulence–transmission relationships include the 'sit-and-wait' [76,77] and the 'Curse of the Pharaoh' hypotheses [78], which state that extended persistence in the off-host environment releases parasites from the constraint of host survival, allowing the evolution of high virulence. Many theoretical studies have explored evolutionary relationships between virulence and parasite longevity [70,74,78–82] where virulence is more likely to increase when its evolution is independent of free-living survival or other transmission traits.

Most empirical tests of theory using ETPs control for the external environment and standardize the transmission phase, to focus on coevolutionary dynamics between host and parasite. However, contrasting results of these experimental tests highlight how seemingly small methodological differences in experimental design may alter selection on transmission, causing significantly different virulence trajectories, even within the same experimental system (e.g. [83,84]). As a result, Rafaluk et al. [85] suggest avoiding artificial manipulation of transmission in experiments, allowing it to proceed in a manner equivalent to how it would in a natural system. While this is an important consideration, it follows that experimental studies for highly persistent environmental parasites become challenging to conduct in laboratory settings.

A recent meta-analysis of the relationship between virulence and environmental persistence found only eight datasets that met the search criteria [86]—that measured environmental longevity and virulence traits from multiple parasite strains. From those datasets, Rafaluk-Mohr concluded that virulence and persistence may be positively linked for bacterial and fungal parasites, while there may be a trade-off for viral parasites. We compiled a list of 22 studies on ETPs that explicitly considered both virulence and transmission traits (i.e. in-host *and* in-environment traits) when looking for

**Table 1.** Selected studies that investigate virulence–transmission trade-offs for environmentally transmitted parasites. Transmission types include obligate killers, opportunists and shed parasites; study types include empirical (E) or theoretical (T). The list presented here only includes 22 studies that considered both in-host and in-environment traits (and did not treat number of propagules produced as both a virulence and transmission trait). This list was compiled from a Google Scholar search pairing virulence and transmission with terminology or hypotheses describing this mode of transmission (i.e. 'Curse of the Pharaoh' OR 'sit-and-wait' OR 'obligate killer'). However, this search underrepresented work on opportunistic parasites. Lacking a unique search term that would identify these papers, we relied on the review by Brown et al. [1] and considered references to include based on citations in and of this paper. Selected studies were scanned to determine suitability and any references cited within these that seemed relevant were added. This exercise made clear that although many studies address aspects of virulence or transmission for parasites with environmental transmission, few explicitly test for relationships between virulence and transmission traits.

| trans. type | parasite | host | in-host traits studied | in-environment traits studied | virulence–transmission relationship | study type | citation |
|---|---|---|---|---|---|---|---|
| obligate killer | baculovirus (virus) | Lymantria dispar (gypsy moth) | mortality rate, cadaver size, rate of early larval death before infectious particles produced | overwinter persistence/transmission | trade-off | E | Fleming-Davies & Dwyer [87] |
| | baculovirus | L. dispar | speed of kill | transmission rate, variation in transmission rate, decay rate | context-dependent outcomes | E/T | Fleming-Davies et al. [88] |
| | bacteriophage (virus) | Escherichia coli (bacterium) | multiplication rate | decay rate | trade-off | E | De Paepe & Taddei [12] |
| | bacteriophage | E. coli | viral growth rate in cells | survival rate in urea, the length of the survival challenge increased over the evolutionary experiment; genetic adaptations for urea resistance | trade-off | E | Heineman & Brown [89] |
| | Paranosema whitei (microsporidian) | Tribolium castaneum (red flour beetle) | parasite-induced host mortality, spore load | dose; persistence not explicitly tested, but experimental design allowed spores to remain over generations | no relationship; virulence increased, not correlated with spore load | E/T | Rafaluk et al. [82,84] |
| | Pasteuria ramosa (bacterium) | Daphnia magna (crustacean) | infectivity; number of transmission stages produced; parasite-induced host mortality; host cellular response to infection | seasonality; parasite infectiousness (evolved over season) | trade-off | E | Auld et al. [90] |
| | vesicular stomatitis virus | cell culture lines: HeLa, MDCK, BHK | plaque size (virulence trait) | survival after passage through host cell lines | trade-off | E | Ogbunugafor et al. [91] |
| | vesicular stomatitis virus | cell culture line: BHK cells | viral fecundity; viral concentration, plaque size | extracellular survival; temperature-dependence | trade-off | E | Wasik et al. [92] |
| | generic spore-producer | none specified | effect of infection on host fitness; parasite replication rate; two parasite genotypes considered (do not compete in hosts, thus more infections = more spores produced) | parasite competition between hosts (parasite fitness determined by spore shedding by the focal infection relative to the number of spores shed by all infections); spore survival capped | no relationship | T | Lively [93] |

(Continued.)

**Table 1.** (Continued.)

| trans. type | parasite | host | in-host traits studied | in-environment traits studied | virulence–transmission relationship | study type | citation |
|---|---|---|---|---|---|---|---|
| opportunistic | Flavobacterium columnare (bacterium) | fish | parasite-induced host mortality rate (after persistence experiment) | persistence in water; change in colony morphology (linked to virulence) | context-dependent outcomes | E | Sundberg et al. [74] |
| | F. columnare | Oncorhynchus mykiss (rainbow trout) | parasite-induced host mortality rate | nutrient levels in environment; expression of virulence genes | none noted; growth in high-nutrient environment increased virulence | E | Penttinen et al. [94] |
| | F. columnare | O. mykiss | parasite-induced host mortality rate | competition; growth in low- or high-nutrient conditions | no relationship | E | Pulkkinen et al. [95] |
| | Serratia marcescens (bacterium) | Galleria mellonella (wax moth) | parasite-induced host mortality rate | parasite–predator coevolutionary experiment; anti-predator defences (a form of environmental survival) including pigmentation, biofilm, growth rate and motility | trade-off | E | Mikonranta et al. [96] |
| | S. marcescens | Drosophila melanogaster (fruit fly) | parasite-induced host mortality rate, pathogen load, protease activity, motility | growth rate | no relationship; virulence decreased when selection on transmission relaxed | E | Mikonranta et al. [97] |
| shedding | pepper mild mottle (virus) | peppers (plants) | resistance-breaking mutations and effects on infectivity, multiplication rate, virulence | particle stability (persistence proxy) | no relationship | E | Bera et al. [98] |
| | none specified | none specified | parasite-induced host mortality rate | propagule decay rate (1/parasite-induced host mortality rate); competition between two parasites | context-dependent outcomes | T | Bonhoeffer et al. [77] |
| | Ophryocystis elektroscirrha (protist) | Danaus plexippus (monarch butterfly) | parasite replication (spore abundance); effect of infection on host survival to adult stage, mating success, lifespan and fecundity | proportion of offspring infected (shed from mother to egg mass) | trade-off | E | de Roode et al. [99] |
| | Glugoides intestinalis (fungus) | D. magna | parasite-induced host mortality rate; in-host growth rate; time to kill; number of infections per cell; per spore transmissibility of the parasite | experiments applying high or low background host mortality; size of sporophorous vesicles | trade-off | E | Ebert & Mangin [100] |
| | none specified | none specified | parasite-induced host mortality rate; parasite competition through host coinfection | cost of dispersal; probability of survival | context-dependent outcomes | T | Gandon [78] |
| | none specified | none specified | variation in growth rate (generation of growth mutants with increased replication in hosts) | variation in survival rate (generation of transmission mutants with enhanced off-host survival) | trade-off | T | Handel & Bennett [101] |
| shedding and obligate killer | none specified | none specified | parasite-induced host mortality, probability of infection, parasite reproduction rate | decay rate | positive relationship under obligate killer; trade-off under shedding | T | Caraco & Wang [80] |
| | none specified | none specified | parasite-induced host mortality rate | decay rate of propagules; spatially structured transmission | context-dependent outcomes | T | Kamo & Boots [79] |

relationships between the two (table 1) These studies show no common relationships between virulence and transmission, neither in general, nor within transmission types (obligate killers, shedders or opportunists), and represent a wide variety of parasite and host taxa, with both experimental and theoretical studies. Our list differs from Rafaluk-Mohr's [86] meta-analysis in the studies included, but similarly notes how few studies have tested this relationship for ETPs, and how mixed the results are.

Despite considerable interest in how parasite persistence and virulence relate to each other, the strongest empirical evidence testing these hypotheses for ETPs comes from comparisons among, not within, parasite species. Human respiratory viral and bacterial infections show a strong positive correlation between case fatality and persistence, among those whose environmental persistence varies from less than a day (e.g. *Haemophilus influenzae*) to years (e.g. variola virus) [76]. By contrast, among bacteriophages that infect *Escherichia coli*, virion survival in the environment is negatively correlated with multiplication rate in host cells [13]. This trade-off in virulence and survival traits was mechanistically linked with the physical properties of the virion, including capsid thickness and density of the packaged genome, where those that had the highest replication rates in cells (i.e. largest burst size) had the least persistence. While trends across broad taxonomic groups in parasite life-history strategies are certainly of interest, they do not address the more pressing management questions of what drives virulence transmission dynamics in a particular host–parasite system, or how we can manage an infection to lessen its impact on a host population.

One confounding aspect of parasite life-history theory is that different hypotheses have been developed for parasites with direct versus environmental transmission when arguably the only difference between direct and environmental transmission lies in the time scale of the off-host period [102]. These hypotheses describe either end of this time spectrum, and may in part explain why there is such mixed support for parasite life-history theories based on empirical tests [9,64,86,103]. Additional research on a range of direct and environmentally transmitted disease systems that vary in persistence, and where transmission is allowed to proceed naturally, would clarify whether specific evolutionary life-history strategies cluster by broad taxonomic parasite groups, and what effect the selection for increased or decreased environmental persistence has on virulence evolution.

## 3. Approach II: patterns across space: parasite adaptation in heterogeneous environments

The second approach to the study of virulence and transmission relationships leverages heterogeneity in the environment to investigate host–parasite dynamics across populations. This approach recognizes that the abiotic and biotic environment can have a strong effect on host–parasite interactions, resulting in context-dependent virulence and transmission [41–43]. In considering host–parasite interactions, parasite local adaptation tends to occur if parasite gene flow is higher than host gene flow among populations [104,105]. While this may seem counterintuitive, gene flow increases the genetic diversity available for selection, increasing the efficiency of parasite local adaptation. Divergent selection pressure across space can lead to local adaptation [106,107], which can be used to infer the effect of the environment on species interactions. Many of the parasites examined under this approach are also ETPs, mostly of plant or invertebrate hosts (see box 4). This approach leverages the variation observed among natural populations, by sampling across metapopulations, or using host and parasite genotypes collected from different environments and grown under different controlled conditions.

One prominent theory linking coevolutionary dynamics of interacting species across metapopulations is the geographical mosaic theory of coevolution (GMTC) [122,123]. For host–parasite interactions, the GMTC emphasizes that selection pressure on host–parasite interactions will vary based on heterogeneous environmental conditions affecting both host and parasite populations [43]. Together, these effects create a mosaic pattern of coevolutionary hot spots and cold spots across the joint range of the species [124]. Selection mosaics, hot-spot dynamics and trait remixing together lead to parasite (and host) diversity across a heterogeneous landscape that can be strongly affected by the environment.

Parasite genotype by environment interactions are important for maintaining genetic variation in parasites, and highlight how environmental heterogeneity leads to different evolutionary outcomes for the parasite. For example, strains of the fungal parasite *Podosphaera plantaginis* vary in their performance (i.e. infectivity, growth and propagule production) at different temperatures [125], a result of coevolution with its plant host (*Plantago lanceolata*) at different temperatures across the plant–parasite metapopulation [126]. Further work in this host–parasite system finds that infection

prevalence and severity vary over small spatial scales, driven by local infection density, humidity and other environmental factors [127].

Two critical environmental factors supporting genotype by environment interactions are temperature and resource availability [43]. Temperature can affect host and parasite performance and a range of infection traits, as documented in systems such as *Daphnia magna–Pasteuria ramosa* [128,129], *Plantago lanceolata–Podosphaera plantaginis* [125,126], *Cryphonectria parasitica–Cryphonectria hypovirus*-1 [130], *Pseudomonas fluorescens*-bacteriophage [131] and numerous insect–parasite systems [132]. Resource availability affects host condition and immune response, and hence infection dynamics (e.g. *Aedes sierrensis–Lambornella clarki* [133], *Daphnia dentifera–Metschnikowia bicuspidata* [134], *Serinus canaria–Plasmodium relictum* [135], *Osteopilus septentrionalis–Aplectana* sp. [136]). Changes in resource availability can also affect parasite virulence [95,135]. Less work has been done on how temperature or resource availability relate to parasite survival in the off-host environment. However, the temperature can also affect parasite survival, as documented for larvae of the monogenean *Discocotyle sagittate* in fresh water [137] and cercariae of the marine trematode *Maritrema subdolum* after emergence from snail hosts [138].

The findings of studies we have clustered under Approach II raise the question of how an environmentally mediated change in infection traits would, in turn, affect the persistence of parasites in the off-host environment. These study systems are poised to provide answers to the theoretical questions presented under Approach I, using systems that subvert some of the limitations of transmission experiments also noted under Approach I. It is important to remember that many of the studies detailed under Approach II examine host–parasite interactions in the context of environmental variation, and not specifically (i) how environmental variation affects parasite survival traits in the off-host environment nor (ii) how those survival traits are related to infection or virulence traits. However, this focus could easily be examined within such studies by also both examining traits for survival in the off-host environment, and variation in survival as it relates to infection traits across populations. Combining Approaches I and II would fill some of the holes in empirical tests of theory for a diversity of host–parasite systems, as detailed under Approach I.

# 4. Approach III: patterns through time, parasite evolutionary history and phylodynamics

A phylogenetic framework involving the reconstruction of a parasite's evolutionary history from contemporaneous genetic data provides a historical lens through which to examine relationships between host–parasite evolution, the environment and disease traits. While temporal variation in disease phenotypes can be studied through detailed longitudinal field studies, these are financially and logistically challenging to carry out and tend to exclude the role of the environment. The most well-known example of this is the European rabbit (*Oryctolagus cuniculus*)–myxoma virus system, where viral strain evolution over half a century has been associated with changes in host resistance and parasite virulence [24].

Parasite phylogenies are valuable for examining the transmission dynamics of microparasites, particularly for rapidly evolving viruses (but see [139]), where closely related lineages may reflect epidemiological links. They have been widely used for identifying infection sources, quantifying cross-species transmission and estimating rates of parasite spatial spread (e.g. [140]). In the ongoing severe acute respiratory syndrome coronavirus 2 (SARS-CoV-2) pandemic, for example, phylogenetic approaches have been used to examine the evolution and transmission of the virus at both local [141] and global [142] scales. The phylodynamic framework, which is based on the concept that ecological and evolutionary processes interact to shape parasite phylogenies [143], has also enabled the estimation of key disease parameters, such as parasite growth rates and $R_0$ [142,144–146]. In addition, phylogenies of known virulence factors can provide insights into the evolution of parasite virulence, which can inform the development of therapeutic or vaccine targets (e.g. [147]). Because these approaches explicitly consider the evolutionary history of a parasite, they provide a powerful means for evaluating disease dynamics and evolutionary trade-offs.

Most studies on parasite virulence have focused on theoretical or experimental systems; while these studies are informative for identifying mutations associated with the virulence phenotype (e.g. [148]), they often ignore the natural selective pressures shaping the virulence genotype [4]. However, the fields of phylogeography, phylodynamics and landscape genetics have recently been extended to investigate how the environment shapes parasite evolution and transmission [149]. These approaches provide a useful means for evaluating host–parasite disease outcomes, while explicitly accounting for both evolutionary history and environmental variation [150]. For example, phylogeographic models

enable mapping ancestral states of relevant traits onto parasite genealogies to reveal how parasite transmission varies across the landscape and among multiple host species, respectively [140,151,152]. Coupled with data on host mortality, these approaches can provide insight into how virulence evolves in variable environments and host backgrounds. Furthermore, the integration of spatially explicit genetic and environmental datasets in a statistical framework has enabled rigorous testing of hypotheses regarding the underlying environmental predictors of parasite spatial spread [153,154].

Linking data on parasite infection phenotypes and phylogeny can provide insight into relationships between virulence factors (i.e. mutations associated with disease outcomes) and parasite fitness. For example, repeated evolution of mutations conferring virulence (or increased virulence) across divergent outbreaks (i.e. convergent or parallel evolution) may suggest that a mutation has a selective advantage (e.g. [155]). By contrast, those virulence-determining mutations that reduce onward parasite transmission (due to a trade-off between virulence and transmission) could result in dead-end lineages and, thus, only occur at the tips of lineages that then disappear from a phylogeny [4]. It is important to note, however, that phylogenetic patterns offer only a qualitative evaluation of fitness and may be misleading for emerging parasites.

A phylogeny reflects the entire evolutionary history of the parasite sample under investigation. For ETPs, this makes disentangling evolutionary changes occurring in- versus off-host a challenge often requiring experimentation. Therefore, like the other approaches, this is not a silver bullet for elucidating generalities about virulence–transmission relationships, and a multi-faceted framework is necessary.

# 5. Some perspectives on transmission and virulence in natural populations

In the previous sections, we reviewed three research approaches used to study the evolution of virulence and transmission, highlighting how the environment can have important effects on virulence–transmission relationships. Within this environmental context, we focused specifically on parasite survival during the off-host life stage. Perhaps counterintuitively, an off-host perspective allows us to conceptually merge direct and environmental transmission. These two transmission types are not separate processes but occur along a time continuum in the off-host environment [102], and should ideally be described by the same unified theory. This is especially true given that many classic examples of directly transmitted parasites are found to have the capability of persisting in the environment for longer than thought (e.g. the *Mycobacterium tuberculosis* complex [156]), have diversity in environmental niches among strains that can affect their infectiousness or virulence (avian influenza [157]), or may show the potential for an environmental reservoir long thought not possible (e.g. *Y. pestis* [158,159]). In addition, even directly transmitted respiratory infections, considered short lived in the environment, have fascinating mechanisms to increase their survival and dispersal in the off-host stage [160,161].

The urgent need to improve our understanding of parasite virulence evolution has been highlighted by the recent emergence of SARS-CoV-2 coronavirus and its relatives SARS (severe acute respiratory syndrome) and MERS (Middle East respiratory syndrome) [162]. The relationships between virulence, transmission and environment in these viruses and their variants need to be understood [163,164], but the rapidly evolving relationship between transmission and virulence is of great concern in emerging diseases in general. Yet, virulence evolution remains a field in which theoretical and empirical studies are often poorly integrated [4,9,103,165–167]. Reviews of this literature tend to recommend more collaboration across disciplinary boundaries to better link theoretical and empirical studies, and adding more real-world complexity of the infection and transmission processes to theoretical models.

Here, we offer some practical and specific ways to achieve these goals and to facilitate our progress as a research community, through interdisciplinary research, conceptual developments and methodological developments. As a companion to these recommendations, we offer a theoretical framework for how to address virulence–transmission relationships in complex natural systems (figure 1) building off the approaches reviewed above. We then use anthrax in wildlife systems as a case study employing several aspects of this framework (box 5), since *B. anthracis* is an ETP commonly invoked for the evolution of high virulence and high environmental survival.

## 5.1. Interdisciplinary research

Interdisciplinary work depends on people of different specialties and backgrounds understanding each other. For this to happen, the ideal is to communicate complex ideas in the simplest language possible, and clearly define the terms used, especially those, like virulence, that can vary greatly in their definitions

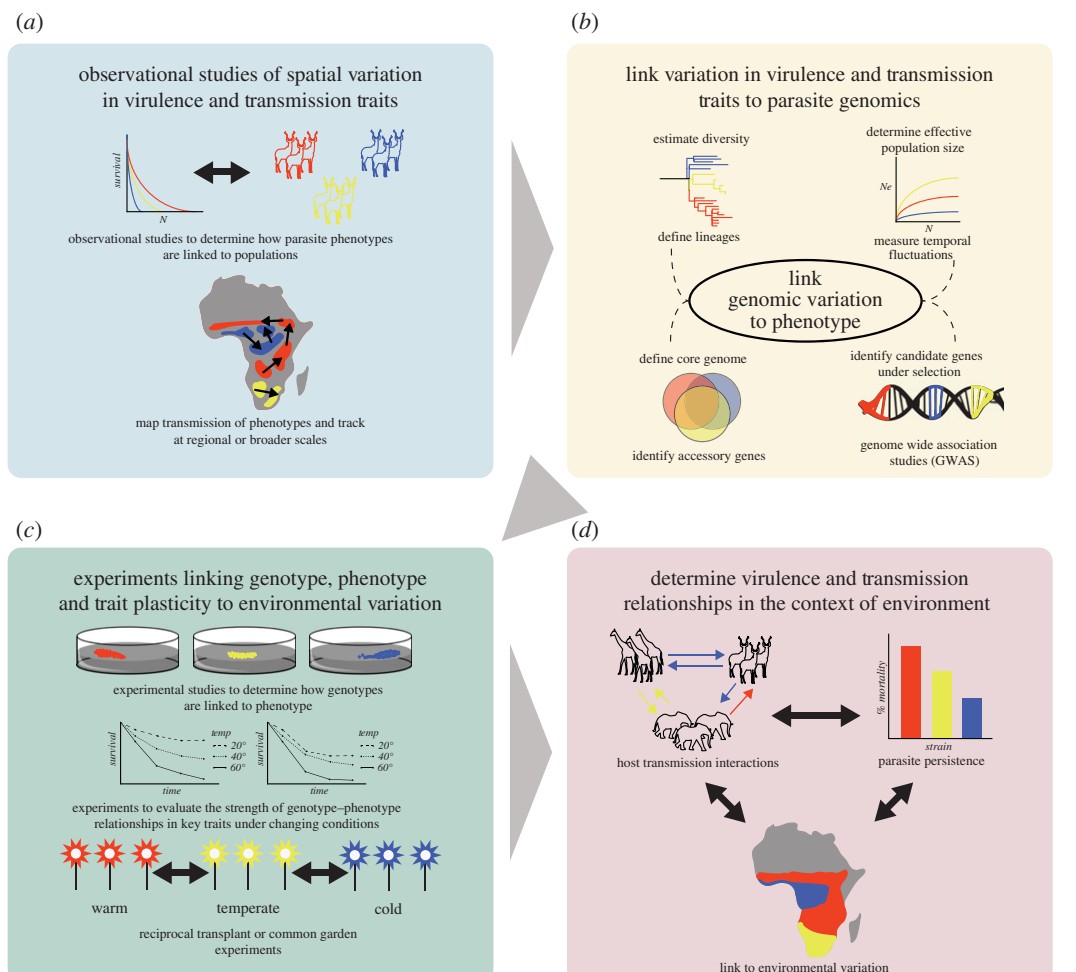

**Figure 1.** A framework for detecting variation in virulence–transmission relationships in natural host–parasite systems. (*a*) From variation detected in observational studies, develop hypotheses of how parasite phenotypes may vary across environmental gradients. Map these traits relative to environmental data to assess putative environmental traits affecting outcomes. (*b*) Use genetic or genomic techniques to link parasite variation of interest to alleles or genes of interest. Use whole genome sequencing to describe the diversity of the parasite across large spatio-temporal scales. Evaluate areas of the genome under selection, and identify putative virulence- and transmission-relevant genes. Link these putative virulence/transmission traits and clades under selection to geography and environmental/spatial variation in (*a*). (*c*) With variable phenotypes identified in (*b*), do controlled experiments to evaluate the strength of genotype–phenotype relationships and the plasticity seen in these traits under different environmental conditions. For systems where experimentation is possible, do reciprocal transplant or common garden experiments to confirm the experimental results from (*c*) in real environments. (*d*) Finally, once genotype–phenotype relationships, and how these vary with environmental variables, are described for the disease system, use that understanding to build statistical and theoretical models of virulence–transmission relationships. These models can test for trade-offs in parasite life-history traits, and determine if there are common environmental factors shaping the outcome of these relationships across space.

across fields. Interdisciplinary work also depends on using data for multiple purposes. A critical aspect here is that the rapidly developing fields of genetics and genomics reconnect with ecology, including ecological data such as time, location, and, when possible, age, sex, condition and phenotype with any gene sequence originating outside a laboratory. This would improve our ability to use data for multiple purposes and draw inference across scales from ecology to genetics and back.

If we rely on tractable laboratory experimental systems to test virulence–transmission theory, we will continue to exclude much of the diversity of taxa involved in host–parasite–environment interactions. Approaches that control for the environment and standardize transmission also limit the evolutionary potential of the system. When possible, evolutionary biologists and microbiologists/parasitologists could team up with field biologists to seek appropriate natural systems and scales in which to test the theory. Useful hypotheses will be reflected in natural systems. For comparative insights over large

**Box 5.** Anthrax as a case study.

Anthrax, caused by the bacterium *Bacillus anthracis*, is often invoked as a textbook example for the evolution of high virulence and high environmental survival [70,78,81,168,169]. Anthrax is an acute and lethal infection [139], with spores that can persist for years in soils [170,171]. Yet, we do not know the variation in these traits, and whether they are truly independent or coupled life-history traits. In part, this is due to anthrax being a reportable and feared disease, making it difficult to study in natural systems. There is also uncertainty in applying a molecular clock to an ETP that spends long periods in a dormant phase, which has inhibited evolutionary studies.

Across its range, anthrax varies in the frequency and severity of outbreaks, and the host species affected [172–174]. Outbreaks vary from sporadic annual cases as observed in Namibian savannahs [175], to explosive outbreaks as observed in the Siberian tundra [176], northern Canada [177] and the Zimbabwean eastern lowveld [174]. In describing its epidemiology, scientists tend to focus on environmental conditions and host ecology, with considerably less work done to explore pathogen ecological or evolutionary interactions (but see [178–181]). *Bacillus anthracis* strains have high genetic similarity globally [182,183], leading to the assumption that strains are also phenotypically monomorphic, and that epidemiological differences must relate to environmental conditions and/or host behaviour, density and susceptibility.

There are, however, intriguing phenotypic differences among strains that could alter their ecological and evolutionary trajectories. The major clades differ in their fitness [182] and *B. anthracis* exhibits local adaptation, or differentiation, of strains in different parts of its range [184–186]. One of the variable-number tandem repeats (VNTRs) used to genotype *B. anthracis* (Ceb-Bams13), located in the *bclA* gene, has been linked to spore phenotype, where repeat length directly correlates with filament length on the exosporium surface [187,188]. The BclA glycoprotein plays a major structural role in the spore coat, and is the site of host recognition by the complement system, initiating phagocytosis and carriage of spores across the epithelium [189]. Laboratory experiments show host species differ in resistance to *B. anthracis*, and that resistance trades off with tolerance to the lethal toxin [190]. These differences among hosts could have adaptive significance to virulence–transmission relationships. However, researchers have yet to test for local adaptation or variation in fitness among *B. anthracis* strains, either in the ability to invade and proliferate in particular hosts, or to persist in the environment.

Ultimately, spatial patterns in strain diversity, distributed across regions varying in soils, climates and host species opens the door for natural selection to act, adapting this pathogen to local conditions. How then can we test for virulence–transmission relationships in such a pathogen? Building off the potential genotype–phenotype variation described above, the next step could compare multiple locations and the ecological and evolutionary interactions between environment, hosts and *B. anthracis* strains. Pathogen genome-wide association studies can link phenotypic variation to putative genetic variation. Once critical traits and genes are identified, small-scale controlled laboratory experiments can test the strength or plasticity in these traits, and their relationship to environmental factors associated with the pathogen's niche (e.g. *in vitro* growth or competition experiments under variable conditions, or spore survival in different soil conditions). Then, we can test mathematical models of transmission and interactions between life-history traits, that may lead to the outcomes observed in different disease systems. Ideally, for anthrax or other host–parasite systems, detecting common drivers of the variation in host–parasite relationships across environmental gradients will allow us to infer properties of these systems that scale beyond the specifics of any particular environment, host species or parasite species.

spatial scales, we can consider the example of other globally distributed experimental research efforts, such as the Nutrient Network [191].

Even the first step of an evolutionary analysis may involve interdisciplinary collaboration for critical examination of each step in the transmission process, including the habitats in which propagules are released, their survival and development in different habitats, and how their spatial distribution affects host contact, transmission pathways and thus selection. Pulling open the lid on the transmission black box allows critical examination of steps in the transmission process, and how these influence disease dynamics and parasite evolution (e.g. [192]).

## 5.2. Conceptual developments

Disease transmission via persistent propagules in the environment is a transmission strategy used by a diversity of parasitic life and the mechanisms for environmental survival are varied, even within taxonomic groups (e.g. pathogenic bacteria [193]). Yet, we lack a general term to describe transmission from an environmental reservoir to a host. Instead, this type of transmission tends to be described by the specific reservoir (e.g. water-borne, soil-borne, food-borne). The difficulty in defining, let alone measuring, appropriate traits for virulence and transmission inhibits our understanding of these relationships (boxes 2 and 3), and developing a common terminology should be a simple and useful early step. For instance, ETPs can be divided into three general types relevant to virulence–transmission relationships: (i) obligate parasites that shed reproductive stages during host infection (shedders), (ii) obligate parasites that release reproductive stages all at once upon host death (obligate killers), and (iii) facultative or opportunistic organisms that can parasitize hosts under certain conditions (opportunists). These categories highlight where replication occurs (inside or outside of hosts) and the degree of reliance on host health and mobility for transmission (continuous shedding through host morbidity versus release upon host death).

It is important to remember that selection acts on the whole life cycle of a parasite [194] and all of its traits, not just on the handful of traits and life stages most easily measured. Hence, within-host infection dynamics and between-host transmission processes must be seen in combination to say anything useful about the selection pressures towards any particular virulence/transmission trade-off or strategy. This is true not just for ETPs, but across the spectrum of direct to environmental transmission, as well as across transmission modes. Recent insights into parasite evolution emphasize the importance of considering both within-host infection dynamics and between-host transmission processes for predicting virulence evolution and epidemiological dynamics [9,17–19]. Thus, research building off the studies highlighted under Approach II could be a fruitful avenue for future work into virulence–transmission relationships.

Any study of virulence should also acknowledge that virulence is an emergent trait of specific host–parasite interactions, and not something that can be viewed as a trait of only one species or even a pair of interacting species. Instead, it may also be dependent on environmental factors or third-species interactions, such as for opportunistic parasites. Parasites often have alternative strategies along a gradient from mutualism and commensalism to environmental opportunists [1] and specialized and obligate parasites. Virulence and transmission are both context-dependent [195], and virulence should be clearly defined as either excess morbidity or mortality in the host incurred by the parasite in a given environmental context.

When virulence is seen as a strategy and not a maladaptive by-product, it must be remembered that parasites have social interactions: they are not homogeneous or automatically cooperating altruistically with each other. Instead, cooperation in shared costs such as virulence factor production depends on social interactions and evolutionary game theory as in macroscopic organisms, and social interactions will affect virulence [196]. The misleading mode of thinking where the infection is treated as the unit of selection is, fortunately, becoming rare in the literature. As many microparasites have rapid population growth and huge populations, they have a lot of opportunity for variation and selection, and should always be treated as such. Ways to approach this could include simplifying the model to 'multiple infections', or by treating microbes more like we treat evolution and population genetics in macroscopic organisms, just sped up. The rapid increase in computational power is making the latter more and more often feasible.

## 5.3. Methodological developments

We need to reconsider how we prioritize the importance of parasite interactions with macroscopic organisms over microscopic organisms, and how we group parasites for evolutionary analysis based a transmission mode (e.g. environmental, direct, arthropod vector-borne) given that often alternative pathways are possible. In this review, we have noted that direct and environmental represent the same mode of transmission in that both require passage through the environment. As opposed to a host, the environment has no active response and does not evolve due to the presence of a parasite. Many parasites with 'environmental' transmission may be vectored by other microorganisms with which they interact during an environmental phase. Examples include the relationships between copepods and *V. cholerae* [197], or amoebae and bacteria such as *Francisella tularensis* [198], *Legionella pneumophila* [199], *Y. pestis* [200] and *B. anthracis* [178]. Thus, we may be biased by instinctively considering macroscopic secondary hosts, vectors and interactions in a different evolutionary light than non-mammalian 'secondary hosts' and non-arthropod 'vectors' where the mutualists/antagonists are chiefly other microorganisms. Metagenomics and other high-throughput methods for determining microbial community composition

may be crucial in addressing an often overlooked aspect of host–parasite interactions—that many parasites have relationships with other microorganisms during their environmental phase.

Comparative studies with data collected across large spatial scales may be needed to detect traits relevant to virulence–transmission relationships among locations, or sources of variation driving virulence–transmission relationships that can then be isolated and tested in controlled laboratory settings (figure 1). Large-scale studies have become more feasible with the integration of nested models and more advanced statistical analysis combined with remotely sensed environmental data on multiple scales over large areas. Research at multiple natural scales will promote an understanding of in-host versus between-host processes in natural systems, a connection that is widely acknowledged as under-studied in the field and laboratory. Further, observational studies of natural host–parasite systems reflect the outcome of evolution unconstrained by what is practically feasible due to time constraints, spatial scales, multispecies interactions or for parasites that are not amenable to laboratory study for various reasons [85].

Much work has been done in recent years to describe and predict the distribution of diseases using ecological niche models [49]. These aim to detect strong effects of environmental variables on the occurrence of diseases, where the environment often is the limiting factor for whether biotic interactions occur, and risk maps can be created with purely environmental variables [49]. However, these models can suffer from a foreshortened time perspective, confounding effects of management, data gaps and confusing the environmental variables that are available (such as temperature or remotely sensed data) with those that are necessary (such as the distribution of hosts, vectors or commensal microbiota). Common examples are climate envelope type models where the 'climate niche' of a disease is inferred from its current distribution. In some cases, this may be valid, but in many instances, they can be wrong due to reporting biases or historical control efforts affecting the recorded distribution. For instance, because anthrax has been strictly managed and controlled in Siberia, with annual large-scale livestock vaccination programmes over decades, there are very few reported modern cases of anthrax [201]. A recent paper using modern cases to infer a 'climate niche' for anthrax thus finds Siberia to be of 'low suitability and low probability of occurrence' [202], despite the fact that these control measures have been in place since Tsarist times for precisely the *opposite* reason. Indeed, the Russian name for anthrax is 'сибирская язва', or 'Siberian plague' due to its historical prevalence in just this area. Thus, while these models are tempting for computationally oriented biologists to use, they need ground-truthing and integration with system-specific knowledge and mechanistic understanding to be of value for management, risk analysis or further understanding of a system.

Genome-wide association studies (GWAS) can be used to detect genes linked with variation in virulence or transmission in parasites [203,204]. GWAS has been used to assess parasite virulence factors [205,206], niche selection [207] and even to detect an evolutionary trade-off between strain toxicity and transmissibility in *Staphylococcus aureus* [208]. Parasite (and host) genomics and phylogenetics across populations or ecosystems can be used to investigate how coevolution proceeds in systems given a range of selection pressures imposed by the host(s), parasites and environment. Combining phylogenetic approaches with GWAS facilitates the detection of putative virulence and transmission traits, and their variation, in natural systems (figure 1). Phylogeographic studies can then assess how parasite dynamics change over time, comparing systems under different selection pressure. By combining these approaches, one may determine the step-wise evolutionary mechanisms that would lead to patterns in the data observed through parasite genetic sequences. Once key variations in parasite genotype–phenotype are detected across spatial scales, the flexibility within these traits can be investigated in experimental studies to detect reaction norms and the phenotypic plasticity of different genotypes for transmission and virulence over a range of environmental conditions.

# 6. Summary

Despite decades of research into virulence–transmission relationships, we have little consensus on the directionality of how and if virulence and transmission are related, especially for transmission modes beyond direct transmission. Thus, we propose that we need not just new tools and new research, but rather new ways of framing this question, refining and defining it to better capture the mechanisms and forces directing the evolution of virulence and transmission, in particular with regard to the role of the environment in these processes (box 6). These innovations are critically important for understanding the dynamics of host–parasite interactions. We are confronting emerging infectious diseases with rapid evolution taking place in novel hosts, and altered temperature and precipitation

**Box 6.** Future questions.

We highlight the following questions as important areas for future research in virulence–transmission relationships. These areas of investigation specifically address how the environment contributes to or alters virulence–transmission relationships between parasites and their hosts:

— How do environmental conditions contribute to the evolution of parasite traits or life-history strategies? Which conditions are most important, and are they predictable?

— How does varying selection pressure along environmental gradients affect parasite environmental survival? Do gradients of variation in parasite survival link to predictable changes in parasite virulence?

— How do species interactions in the off-host phase alter disease dynamics (e.g. the role of microbial predators or competitors on parasite density and infectiousness)?

— How strong is the link between parasite genetic traits and phenotypic traits that are relevant to life-history trade-offs? To what degree does phenotypic plasticity (or epigenetics) enable parasite success under a range of conditions in the external environment? Does the amount of phenotypic plasticity possible vary predictably among parasite taxa?

— Can we leverage the variation in environmental conditions across a parasite's range to understand how conditions in the environment alter virulence and transmission traits, and host–parasite dynamics? Can we then predict the epidemiological outcomes of parasite emergence in novel geographical regions?

— How does context-dependent selection alter virulence and transmission traits across a parasite's range? How do these processes compare for mobile versus stationary hosts? For different parasite taxa? For different transmission modes or routes?

patterns as a result of land use transformations and global climate change. With better understanding of how the environment alters disease dynamics, we can enhance our abilities to predict or mitigate disease outbreaks in a changing environment.

Data accessibility. This article has no additional data.

Authors' contributions. W.C.T. and K.K. conceived of the study. W.C.T., Y.-H.H. and Z.R.B. performed the literature review, W.C.T., K.K. and P.L.K. wrote the first draft of the manuscript, S.A.B. made the figure, and all authors contributed to revisions.

Competing interests. We declare we have no competing interests.

Funding. This perspective was inspired by our research funded under NSF grant no. DEB-1816161/DEB-2106221.

Acknowledgements. For comments on some of the many previous versions of this manuscript, we thank Thomas Caraco, Ing-Nang Wang, Wolfgang Beyer, the Ecology, Evolution and Everything seminar group at the University of Maine, and insightful anonymous reviewers on this submission and a previous version of this manuscript.

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
