## [Peer Review File · Royal Society Open Science]

Review History

RSOS-210088.R0 (Original submission)

Review form: Reviewer 1

Is the manuscript scientifically sound in its present form?

Yes

Are the interpretations and conclusions justified by the results?

Yes

Is the language acceptable?

Yes

Do you have any ethical concerns with this paper?

No

Have you any concerns about statistical analyses in this paper?

No

Recommendation?

Accept with minor revision (please list in comments)

Comments to the Author(s)

Please see attached PDF document (Appendix A).

Review form: Reviewer 2

Is the manuscript scientifically sound in its present form?

Yes

Are the interpretations and conclusions justified by the results?

Yes

Is the language acceptable?

Yes

Do you have any ethical concerns with this paper?

No

Have you any concerns about statistical analyses in this paper?

No

Recommendation?

Accept with minor revision (please list in comments)

Comments to the Author(s)**GENERAL COMMENTS:**

This paper is a review of research on transmission-virulence relationships, presented through the perspective that considering the effect of environmental persistence (the off-host stage) of the parasite can clarify existing inconsistencies and lead to novel and robust insights. The authors frame the existing literature based on three research approaches: studies of virulence-transmission trade-offs, studies of local adaptation, and studies using phylodynamics, and then discuss how these and other approaches can be integrated to yield new insights into disease dynamics and transmission-virulence relationships.

The first half of the paper does a nice job reviewing virulence-transmission literature within the framework of the three common approaches. The organization of research based on the three approaches is useful. The writing is clear and easy to follow. The thesis that environmental persistence must be considered in virulence-transmission trade-offs is important and worthwhile. These strengths argue for the value of this manuscript. However, while the thesis is clearly stated in the introduction and abstract, some sections blurred that point or seemed to conflate it with other arguments. For example, the importance of considering the effect of the environment on parasite traits is sometimes blurred with the importance of studying environmentally-transmitted parasites. Similarly, virulence-transmission and virulence-persistence seem to be used synonymously in a section that discusses how persistence affects virulence-transmission relationships. (Therefore the use of terms is circular). See details below.

The authors propose integration of the three approaches to achieve clearer and more consistent results regarding transmission-virulence trade-offs (Figure 1). For example, findings from Approach II (effects of environment on parasite traits or host-parasite dynamics) should be

integrated into Approach I for a more robust and consistent understanding of virulence-transmission relationships. This is a useful framework overall. However, the final step in this framework (D in Figure 1) is a bit vague. Admittedly, this vagueness could stem from the fact that this final step depends on the specifics of the study system. However, in that case the authors can, and I argue should, provide a concrete example of this last step in their case study of Anthrax (Box 5). As is, the case study provides only a general statement that the results of the previous steps in the study would be used to “build models”.

The second half of the paper (“Some perspectives on transmission and virulence in natural populations”) is informative. However, it is a bit of a departure from the first half of the paper and the thesis of considering environmentally transmitted parasites (ETPs). It covers a very broad spectrum of recommendations, including the need for interdisciplinary research, new conceptual developments, and new methods (e.g. metagenomics, studies at large spatial scales, improvements to niche models, and genome-wide association studies.) Again, this is useful, it just seems extremely broad and somewhat disconnected from the thesis and structure of the first half of the paper.

Overall, I found this to be an interesting and informative review. The argument that we need to consider ETPs, and to view them on a continuum rather than as fundamentally different from directly transmitted parasites, is broadly important to the study of disease dynamics and evolution and is well substantiated within the manuscript. My critiques are mostly aimed at tightening and clarifying the manuscript. I might also suggest breaking the paper into two articles based on the weaker connection between the first and second half - however, this issue is partly a matter of preference regarding the organization of review articles, and should be left up to the editor and authors to assess.

SPECIFIC COMMENTS:

1. Figure 1. What is a “virulence-transmission” trait? The conceptual figure shows the example of the shape of the curve of N vs survival. However, it’s hard to imagine how this is a directly measurable trait. I also can’t think of any other “virulence-transmission” traits. Do you mean virulence traits and transmission traits, as separate entities? Please clarify.

2. Section “Approach I” argues that studies examining virulence-transmission trade-offs often fail to consider the effect of environmental variation on the virulence-transmission relationship. By controlling transmission, these studies miss the effect of environment. This section was generally well written and informative, but two points need clarification:

2a. It was unclear how studies “control transmission”. Please specify what you mean.

2b. The section seems to conflate the concept of a transmission-virulence relationship with a persistence-virulence relationship (e.g. Line 110 vs Line 152). This is problematic because, as I understand it, a central point of the manuscript is that environmental persistence AFFECTS the transmission-virulence relationship, so transmission and persistence cannot be synonymous.

3. Section “Approach II” reviews studies of local adaptation or geographic variation in parasite traits or host-parasite dynamics. The authors conclude that such studies of spatial variation should be used to inform transmission-virulence studies (Approach I). Approach I would be strengthened by considering how environmental variation affects parasite traits and transmission. This section is good and I have only one comment here:

3a. This section switches between discussion of two topics (1) environmental effects on traits and (2) environmental persistence. The distinction between these two is at times blurred and confusing.

3b. L. 231-235. This statement suggests that the whole section (Approach II) is about host-parasite co-evolution (geographic mosaic theory), not parasite traits. I disagree: some of the statements in this section discuss traits such as virulence, growth, or infectivity.

4. Section "Approach II" reviews phylodynamic approaches to inferring disease dynamics, population size, or transmission. Studies of functional genes can detect evolution of virulence determinants. The authors emphasize that interpretation of phylodynamic studies becomes uncertain with environmentally transmitted parasites (ETPs).

5. BOXES:

The "point" of Box 3 is not clear. It starts by pointing out that R_0 is used oversimplistically and that it's important to consider time. It goes on to note that virulence and transmission are conflated in many studies. It ends by emphasizing the importance (and current neglect) of environmental persistence and the need for a unified theory that sees environmental persistence along a continuum. These are all good points, but the point of the box overall is unclear.

In general, all of the boxes are too long, each containing a full page or more of single-spaced text. Boxes should be concise presentations of concepts or issues relevant to the main body of the manuscript. I recommend condensing the content of each box to clearly and efficiently convey their points.

6. Table 1: This is an informative Table. However, what is meant by "alternate strategies evolved" under "Virulence-transmission relationship"?

7. L. 392. change "may be vectored by other microbes" to "may be vectored by other organisms"? The first example is of copepods vectoring *V. cholerae*.

8. L. 393 *Vibrio cholerae* is misspelled as "Vibrio cholera")

9. L. 1217 Write out Variable Number Tandem Repeats at the first use of the acronym.

Decision letter (RSOS-210088.R0)

Dear Dr Turner

The Editors assigned to your paper RSOS-210088 "How parasite environmental survival affects virulence-transmission relationships" have now received comments from reviewers and would like you to revise the paper in accordance with the reviewer comments and any comments from the Editors. Please note this decision does not guarantee eventual acceptance.

We invite you to respond to the comments supplied below and revise your manuscript. Below the referees' and Editors' comments (where applicable) we provide additional requirements.

Final acceptance of your manuscript is dependent on these requirements being met. We provide guidance below to help you prepare your revision.

Please submit your revised manuscript and required files (see below) no later than 21 days from today's (ie 23-Feb-2021) date. Note: the ScholarOne system will 'lock' if submission of the revision is attempted 21 or more days after the deadline. If you do not think you will be able to meet this deadline please contact the editorial office immediately.

on behalf of Dr Cynthia Downs (Associate Editor) and Pete Smith (Subject Editor)
openscience@royalsociety.org

Associate Editor Comments to Author (Dr Cynthia Downs):

Associate Editor: 1

Comments to the Author:

Two experts in disease ecology and I reviewed this manuscript. Both reviewers praise the clarity and value of the first section of the manuscript, which describes three approaches to virulence-transmission research. The reviewers find that section clear and a valuable summary of the literature. However, both reviewers also note that the review conflates these ideas with the discussion of others, including a substantial discussion of the role of the environment in shaping the off-host stage of environmentally transmitted parasite and an unstated aim of how the three approaches can be integrated. Both reviewers argue that the review lacks some focus because these additional aims are unstated. I encourage the authors to take reviewer 1's advice and to expand/rework the introduction to state all three aims upfront as a mechanism for providing more focus/organization to a well-researched review. I also encourage the authors to give a concrete example for integrating the three approaches in the second half of the manuscript, as suggested by reviewer 2.

Reviewer comments to Author:

Reviewer: 1

Comments to the Author(s)

Please see attached PDF document

Reviewer: 2

Comments to the Author(s)

GENERAL COMMENTS:

This paper is a review of research on transmission-virulence relationships, presented through the perspective that considering the effect of environmental persistence (the off-host stage) of the parasite can clarify existing inconsistencies and lead to novel and robust insights. The authors frame the existing literature based on three research approaches: studies of virulence-transmission trade-offs, studies of local adaptation, and studies using phylodynamics, and then discuss how these and other approaches can be integrated to yield new insights into disease dynamics and transmission-virulence relationships.

The first half of the paper does a nice job reviewing virulence-transmission literature within the framework of the three common approaches. The organization of research based on the three approaches is useful. The writing is clear and easy to follow. The thesis that environmental persistence must be considered in virulence-transmission trade-offs is important and worthwhile. These strengths argue for the value of this manuscript. However, while the thesis is clearly stated in the introduction and abstract, some sections blurred that point or seemed to conflate it with other arguments. For example, the importance of considering the effect of the environment on parasite traits is sometimes blurred with the importance of studying environmentally-transmitted parasites. Similarly, virulence-transmission and virulence-persistence seem to be used synonymously in a section that discusses how persistence affects virulence-transmission relationships. (Therefore the use of terms is circular). See details below.

The authors propose integration of the three approaches to achieve clearer and more consistent results regarding transmission-virulence trade-offs (Figure 1). For example, findings from Approach II (effects of environment on parasite traits or host-parasite dynamics) should be integrated into Approach I for a more robust and consistent understanding of virulence-transmission relationships. This is a useful framework overall. However, the final step in this framework (D in Figure 1) is a bit vague. Admittedly, this vagueness could stem from the fact that this final step depends on the specifics of the study system. However, in that case the authors can, and I argue should, provide a concrete example of this last step in their case study of Anthrax (Box 5). As is, the case study provides only a general statement that the results of the previous steps in the study would be used to “build models”.

The second half of the paper (“Some perspectives on transmission and virulence in natural populations”) is informative. However, it is a bit of a departure from the first half of the paper and the thesis of considering environmentally transmitted parasites (ETPs). It covers a very broad spectrum of recommendations, including the need for interdisciplinary research, new conceptual developments, and new methods (e.g. metagenomics, studies at large spatial scales, improvements to niche models, and genome-wide association studies.) Again, this is useful, it just seems extremely broad and somewhat disconnected from the thesis and structure of the first half of the paper.

Overall, I found this to be an interesting and informative review. The argument that we need to consider ETPs, and to view them on a continuum rather than as fundamentally different from directly transmitted parasites, is broadly important to the study of disease dynamics and evolution and is well substantiated within the manuscript. My critiques are mostly aimed at tightening and clarifying the manuscript. I might also suggest breaking the paper into two articles based on the weaker connection between the first and second half - however, this issue is partly a matter of preference regarding the organization of review articles, and should be left up to the editor and authors to assess.

SPECIFIC COMMENTS:

1. Figure 1. What is a “virulence-transmission” trait? The conceptual figure shows the example of the shape of the curve of N vs survival. However, it’s hard to imagine how this is a directly measurable trait. I also can’t think of any other “virulence-transmission” traits. Do you mean virulence traits and transmission traits, as separate entities? Please clarify.

2. Section “Approach I” argues that studies examining virulence-transmission trade-offs often fail to consider the effect of environmental variation on the virulence-transmission relationship. By controlling transmission, these studies miss the effect of environment. This section was generally well written and informative, but two points need clarification:

2a. It was unclear how studies “control transmission”. Please specify what you mean.

2b. The section seems to conflate the concept of a transmission-virulence relationship with a persistence-virulence relationship (e.g. Line 110 vs Line 152). This is problematic because, as I understand it, a central point of the manuscript is that environmental persistence AFFECTS the transmission-virulence relationship, so transmission and persistence cannot be synonymous.

3. Section “Approach II” reviews studies of local adaptation or geographic variation in parasite traits or host-parasite dynamics. The authors conclude that such studies of spatial variation should be used to inform transmission-virulence studies (Approach I). Approach I would be strengthened by considering how environmental variation affects parasite traits and transmission. This section is good and I have only one comment here:

3a. This section switches between discussion of two topics (1) environmental effects on traits and (2) environmental persistence. The distinction between these two is at times blurred and confusing.

3b. L. 231-235. This statement suggests that the whole section (Approach II) is about host-parasite co-evolution (geographic mosaic theory), not parasite traits. I disagree: some of the statements in this section discuss traits such as virulence, growth, or infectivity.

4. Section “Approach II” reviews phylodynamic approaches to inferring disease dynamics, population size, or transmission. Studies of functional genes can detect evolution of virulence determinants. The authors emphasize that interpretation of phylodynamic studies becomes uncertain with environmentally transmitted parasites (ETPs).

5. BOXES:

The “point” of Box 3 is not clear. It starts by pointing out that R_0 is used oversimplistically and that it’s important to consider time. It goes on to note that virulence and transmission are conflated in many studies. It ends by emphasizing the importance (and current neglect) of environmental persistence and the need for a unified theory that sees environmental persistence along a continuum. These are all good points, but the point of the box overall is unclear.

In general, all of the boxes are too long, each containing a full page or more of single-spaced text. Boxes should be concise presentations of concepts or issues relevant to the main body of the manuscript. I recommend condensing the content of each box to clearly and efficiently convey their points.

6. Table 1: This is an informative Table. However, what is meant by “alternate strategies evolved” under “Virulence-transmission relationship”?

7. L. 392. change “may be vectored by other microbes” to “may be vectored by other organisms”? The first example is of copepods vectoring *V. cholerae*.

8. L. 393 *Vibrio cholerae* is misspelled as “*Vibrio cholera*”)

9. L. 1217 Write out Variable Number Tandem Repeats at the first use of the acronym.

===PREPARING YOUR MANUSCRIPT===

===PREPARING YOUR REVISION IN SCHOLARONE===

Author's Response to Decision Letter for (RSOS-210088.R0)

See Appendix B.

RSOS-210088.R1 (Revision)

Review form: Reviewer 1

Is the manuscript scientifically sound in its present form?

Yes

Are the interpretations and conclusions justified by the results?

Yes

Is the language acceptable?

Yes

Do you have any ethical concerns with this paper?

No

Have you any concerns about statistical analyses in this paper?

No

Recommendation?

Accept as is

Comments to the Author(s)

I previously reviewed the earlier version of this manuscript submitted to RSOS. My main suggestion was that the authors should improve the manuscript by adjusting the stated focus and being more explicit about each of the intended objectives of the manuscript. I also noted a discontinuity between the objectives as they were originally described in the abstract and the introduction, and the ultimate direction and flow of the manuscript later in the discussion and perspectives sections. The authors have made clear efforts to address these concerns revising their manuscript, particularly in the abstract and introduction to frame their manuscript differently to better justify the scope and content of following sections. I now find that the overall picture of what the manuscript is trying to achieve is much clearer and consistent throughout. Having read the author responses to my other previous comments, the comments of the second reviewer, and the revised manuscript, I think the authors have adequately addressed previous concerns. I have no more additional concerns or comments about the manuscript, and cannot make any further suggestions that would substantially improve the manuscript further. I would therefore recommend acceptance for publication.

Typos / Line specific comments:

L81 Missing space to start sentence "This is not only..."

L230 double space "of the fungal"

Review form: Reviewer 2

Is the manuscript scientifically sound in its present form?

Yes

Are the interpretations and conclusions justified by the results?

Yes

Is the language acceptable?

Yes

Do you have any ethical concerns with this paper?

No

Have you any concerns about statistical analyses in this paper?

No

Recommendation?

Accept as is

Comments to the Author(s)

I reviewed a previous version of this manuscript. I provided comments aimed at tightening the focus of the manuscript, which had previously explicitly presented one objective (the importance of understanding off-host stages of parasites) but actually discussed others as well (virulence-transmission relationships and the role of the environment in shaping host-pathogen dynamics and evolution.). The authors have made revisions that make clear the multiple objectives of the paper and how they fit together. They have done this mainly by (1) adding text to explicitly present the three objectives, (2) making revisions for clarity, (3) making revisions to the Boxes: in particular Box 2 is now more clearly focused. Box 5 gives a somewhat more concrete example of the proposed integrated research process – still not as concrete as I would have liked but sufficient to enable readers to envision the process well enough. One new comment: I would specify “environmental survival” (not just “survival”) in the title. As I noted in my initial review, this is an interesting and thoughtful review that provides a perspective deserving of explicit attention. The authors have made the necessary revisions to address my concerns and I think this paper is ready for publication.

Decision letter (RSOS-210088.R1)

Dear Dr Turner,

It is a pleasure to accept your manuscript entitled "The roles of environmental variation and parasite survival in virulence-transmission relationships" in its current form for publication in Royal Society Open Science. The comments of the reviewer(s) who reviewed your manuscript are included at the foot of this letter.

You can expect to receive a proof of your article in the near future. Please contact the editorial office (openscience@royalsociety.org) and the production office (openscience_proofs@royalsociety.org) to let us know if you are likely to be away from e-mail

contact – if you are going to be away, please nominate a co-author (if available) to manage the proofing process, and ensure they are copied into your email to the journal.

on behalf of Dr Cynthia Downs (Associate Editor) and Pete Smith (Subject Editor)
openscience@royalsociety.org

Associate Editor Comments to Author (Dr Cynthia Downs):

Comments to the Author:

The two original reviewers and I reviewed this manuscript. I enjoyed reading this revised version, and thank you for taking the time to address the reviewer comments thoughtfully. In particular, I appreciate the inclusion of a glossary to help clarify terms used within this manuscript and the integration of the two halves of the manuscript to improve clarity and consistency. Both reviewers and I are satisfied by the changes and have no additional significant comments. I'm sure this manuscript was a considerable undertaking, and I think that the end product will spark thoughtful conversations.

Line comment:

Line 378: I suggest replacing "that cuts across" with "utilized by a."

Reviewer comments to Author:

Reviewer: 1

Comments to the Author(s)

I previously reviewed the earlier version of this manuscript submitted to RSOS. My main suggestion was that the authors should improve the manuscript by adjusting the stated focus and being more explicit about each of the intended objectives of the manuscript. I also noted a discontinuity between the objectives as they were originally described in the abstract and the introduction, and the ultimate direction and flow of the manuscript later in the discussion and perspectives sections. The authors have made clear efforts to address these concerns revising their manuscript, particularly in the abstract and introduction to frame their manuscript differently to better justify the scope and content of following sections. I now find that the overall picture of what the manuscript is trying to achieve is much clearer and consistent throughout. Having read the author responses to my other previous comments, the comments of the second reviewer, and the revised manuscript, I think the authors have adequately addressed previous concerns. I have no more additional concerns or comments about the manuscript, and cannot

make any further suggestions that would substantially improve the manuscript further. I would therefore recommend acceptance for publication.

Typos / Line specific comments:

L81 Missing space to start sentence "This is not only..."

L230 double space "of the fungal"

Reviewer: 2

Comments to the Author(s)

I reviewed a previous version of this manuscript. I provided comments aimed at tightening the focus of the manuscript, which had previously explicitly presented one objective (the importance of understanding off-host stages of parasites) but actually discussed others as well (virulence-transmission relationships and the role of the environment in shaping host-pathogen dynamics and evolution.). The authors have made revisions that make clear the multiple objectives of the paper and how they fit together. They have done this mainly by (1) adding text to explicitly present the three objectives, (2) making revisions for clarity, (3) making revisions to the Boxes: in particular Box 2 is now more clearly focused. Box 5 gives a somewhat more concrete example of the proposed integrated research process – still not as concrete as I would have liked but sufficient to enable readers to envision the process well enough. One new comment: I would specify "environmental survival" (not just "survival") in the title. As I noted in my initial review, this is an interesting and thoughtful review that provides a perspective deserving of explicit attention. The authors have made the necessary revisions to address my concerns and I think this paper is ready for publication.

Appendix A

This manuscript explores and reviews existing literature on host-parasite-environment interactions. The literature is grouped into three broad complementary approaches that capture the motivation and interests of biological disciplines that have studied host-parasite dynamics to date. The authors call for greater consideration of environmental factors that influence host-parasite dynamics, particularly with a view to better resolve broadscale evolutionary patterns in virulence and transmission of parasites, which until now have been remained unclear and even conflicting. The authors emphasise that the environment can especially affect off-host parasite stages, which are rarely considered, but are also subject to selective pressures which will in turn affect host-parasite dynamics, and relationships between virulence and transmission. The authors also highlight the value of integrating the three approaches they have identified better incorporate the role of the environment in future studies of host-parasite dynamics. The authors suggest ways how future research can be approached integratively and suggest clear and appropriate questions that could be addressed going forward.

On the whole I think this manuscript is a very good overview of the literature that has clearly taken a great deal of research and work to compile. I enjoyed reading this manuscript and learnt a few things!

General Comments

My one major comment is to question precisely where the intended focus or aim of the review is? To me, there seems to be three concurrent focuses/aims of the manuscript 1) the off-host part of the life-cycle, 2) the role of the environment in interaction with host-parasite dynamics generally, including in-host (e.g. L456 and Box 6), and 3) the three different approaches that can be integrated to better address questions in future research and advance our understanding of parasitism. Only 1) is stated explicitly by the authors as the focus of the review on L35 and L55, however it seems like more attention is paid to 2) the role of environmental factors more generally and in-host. Focus 3) is clearly mentioned more as a lens for considering the literature but I think is really part of the intended message to integrate disciplines and direct future research. I think all three of these aims can be, and are, achieved with the manuscript. I just feel like what I expected after reading the abstract and the first paragraph of the introduction, was not exactly the same direction the manuscript took later on, particularly in the perspectives section.

I think the way forward is relatively simple and not much is required to address this. I think an addition or two to the introduction and/or some changes of phrasing and emphasis would achieve this clarity of focus, rather than any changes of content:

Aims 2), and especially 3) are not limited to the off-host part of the life-cycle, so it may be better to explicitly state that these are additional aims that would benefit studies of host-parasite dynamics more broadly – perhaps even in the abstract to set the reader's expectation.

A great deal of literature is explored to demonstrate the importance of environmental effects on host-parasite dynamics generally, and not just specifically during off-host stages. I initially thought the authors were perhaps conflating these general environmental effects with off-host process. The authors do later clearly acknowledge this distinction L231-237, but I feel that this acknowledgment should come sooner and be stated as part of the introduction for clarity, perhaps along with an explanation why environmental effects are explored at length, including in-host. As I interpret it, the authors do this because they are exploring broader literature that addresses environmental effects at any life stage (through the 3 approaches) to highlight how the environment might plausibly affect specifically the off-host life stages – which has been understudied in the context of virulence and

transmission (table 1). This set-up broadly speaking – to consider the environmental effects on off-host stages as an extended case of environmental effects generally – is also then a good primer for considering the off-host part of the life cycle as the long-arm of the same temporal spectrum between direct transmission and environmental transmission, where the environment affects host-parasite dynamics across all life stages, either in- or off-host (L174-177).

I hope that I have articulated my thoughts clearly. My line-specific comments were (more or less) written at first read, but I hope they indicate the points at which the focus was unclear to me and could be differently emphasised.

Line-specific Comments

Abstract

L43 This last sentence is slightly disconnected to the focus of off-host stages? This is a much broader application beyond off-host.

Introduction:

L55 Could possibly explicitly state here something like “While the environment can influence host-parasite dynamics at any stage, here we extend this focus...”

L61-63 Virulence as a composite trait of both the parasite and the host?

L81-89 This paragraph started broadly but limited environmental effects to climate and did not explore further biotic factors or non-climatic abiotic factors off-host. I think each existing sentence is appropriate here, but could be expanded upon slightly for a complete entry into thinking about the off-host environment. Perhaps an example of a biotic interaction outside the host that would affect virulence and/or transmission?

L84-86 [separate point] These environmental factors are not necessarily affecting off-host life stages though? For example, ref [40], using climate data to predict Dengue outbreaks, but Dengue doesn't have an off-host stage? I think the climate variables here better predict availability of free-standing water required for mosquito reproduction, and consequently Dengue vertical transmission and proliferation in mosquitoes. If the focus of this review as stated on L55-57 is the off-host portion of the parasite life-cycle, then would it be better to use examples from parasites with a free-living, or motile life stages, or environmentally persistent propagules, if used in this context? Otherwise, is this conflating environmental factors that affect all life-stages with environmental factors that affect off-host stages? [This comment was written before reading L231-237; perhaps somewhere at the start of this paragraph or at L55 it would work to explain that you explore environmental effects on host-parasite systems generally, both in- and off-host in this review to understand the diversity of environmental impacts that could theoretically impact the off-host stages]

Approach 1

L145 Rafaluk and Jansen?

L166-170 Is this potentially a typo with the relationship the wrong way around on L170? “survival in the environment is positively correlated with multiplication rate in host cells” and in the following sentence “the highest replication rates in cells (i.e. largest burst size) had the least persistence”; these seem to directly contradict one another? Is there some technical difference between multiplication and replication rates and an understanding of the burst size is important to this point?

L182 This should be 'effect' not 'affect'.

Approach 2

L185 Should the comma in this section title be a colon or semi-colon. It doesn't quite read right with a comma.

L189-191 For me, this sentence seemed a little counter intuitive as gene flow usually results in genetic homogeneity among populations. Having now read ref [71] to better understand, it could be good to add something explanatory to this sentence i.e. that gene flow actually increases genetic diversity available for selection which increases the efficiency of local adaptation.

L222-223 Again, the off-host environment is stated as the main focus of the manuscript but is only mentioned here briefly in discussion of the larger environmental context that shapes host-parasite dynamics. I think what is written here is fine, but the emphasis earlier in the manuscript should change. L193

L231-237 This is the important acknowledgement that environmental effects on the whole host-parasite system and environmental off-host effects are not the same thing. Perhaps this critical point/framework should come much earlier in the introduction as it helps avoid possible conflation of the environmental effect as a whole and in off-host stages. Doing so would also help to explain why you explore some in-host environmental effects at length, while the focus of the paper is really stated as the off-host environmental effects. This paragraph could then remain, but in a pared down form that simply describes the benefits of combining approaches 1 and 2.

Approach 3

No comments

Perspectives

L305-309 Perhaps here you should state explicitly that anthrax is a good example because it is an ETP with a highly persistent off-host life-cycle stage to keep with the off-host focus. [You do this in the Box]

L340-343 These two sentences seem contradictory. The first sentence here seems to infer missing terminology, but the second sentence seems to describe terminology that is commonly used for apparently this purpose. Why is this terminology inadequate for the purpose described, or is it being suggested here that this terminology should be adopted consistently and with the definition that e.g. an ETP that persists specifically in water before transmission to a host should be consistently be called a water-borne parasite, and 'water-borne' should be a term reserved only for these types of parasites?

L355 'between-host transmission processes...' Would it be worth being explicit here that this includes everything on the spectrum from direct transmission to extended periods off-host?

L395-398 This sentence was hard to digest. Could the same point be simplified/split into two sentences?

L391-392 Is this what was meant by biotic environmental factors in the introduction at L83-84? Could this be mentioned briefly in the introduction?

Boxes

L1051 It could be worth qualifying this and stating something like ‘Examples include many respiratory infections spread through coughing and sneezing etc., as well as...’. I didn’t immediately make the link between shedding and respiratory infections collectively.

L1217 Could VNTR be lengthened to variable-number tandem repeat (?) as this is its first and only usage.

Table 1: In the column headings would it not make more sense to have “Parasite” rather than “Pathogen” given the broad sense interpretation of Parasite throughout the paper?

Appendix B

Reviewer/Editor comments shown in black font, our responses in blue.

Dear Dr Turner

The Editors assigned to your paper RSOS-210088 "How parasite environmental survival affects virulence-transmission relationships" have now received comments from reviewers and would like you to revise the paper in accordance with the reviewer comments and any comments from the Editors. Please note this decision does not guarantee eventual acceptance.

Please submit your revised manuscript and required files (see below) no later than 21 days from today's (ie 23-Feb-2021) date. Note: the ScholarOne system will 'lock' if submission of the revision is attempted 21 or more days after the deadline. If you do not think you will be able to meet this deadline please contact the editorial office immediately.

on behalf of Dr Cynthia Downs (Associate Editor) and Pete Smith (Subject Editor)
openscience@royalsociety.org

Associate Editor Comments to Author (Dr Cynthia Downs):

Associate Editor: 1

Comments to the Author:

Two experts in disease ecology and I reviewed this manuscript. Both reviewers praise the clarity and value of the first section of the manuscript, which describes three approaches to virulence-transmission research. The reviewers find that section clear and a valuable summary of the literature. However, both reviewers also note that the review conflates these ideas with the discussion of others, including a substantial discussion of the role of the environment in shaping the off-host stage of environmentally transmitted parasite and an unstated aim of how the three approaches can be integrated. Both reviewers argue that the review lacks some focus because these additional aims are unstated. I encourage the authors to take reviewer 1's advice and to expand/rework the introduction to state all three aims upfront as a mechanism for providing more focus/organization to a well-researched review. I also encourage the authors to give a concrete example for integrating the three approaches in the second half of the manuscript, as suggested by reviewer 2.

Response: We are grateful for the reviewers' and editor's close reading and insightful evaluation of this paper. These reviews were really valuable at catching areas where we were not clear, and at offering helpful suggestions on how to improve the manuscript. We have made additions and edits throughout to weave these comments into the manuscript, and are very happy with the improvements! Specific details are listed below.

Reviewer comments to Author:

Reviewer: 1

This manuscript explores and reviews existing literature on host-parasite-environment interactions. The literature is grouped into three broad complementary approaches that capture the motivation and interests of biological disciplines that have studied host-parasite dynamics to date. The authors call for greater consideration of environmental factors that influence host-parasite dynamics, particularly with a view to better resolve broadscale evolutionary patterns in virulence and transmission of parasites, which until now have been remained unclear and even conflicting. The authors emphasise that the environment can especially affect off-host parasite stages, which are rarely considered, but are also subject to selective pressures which will in turn affect host-parasite dynamics, and relationships between virulence and transmission. The authors also highlight the value of integrating the three approaches they have identified better incorporate the role of the environment in future studies of host-parasite dynamics. The authors suggest ways how future research can be approached integratively and suggest clear and appropriate questions that could be addressed going forward. On the whole I think this manuscript is a very good overview of the literature that has clearly taken a great deal of research and work to compile. I enjoyed reading this manuscript and learnt a few things!

Response: We appreciate these positive comments!

General Comments

My one major comment is to question precisely where the intended focus or aim of the review is? To me, there seems to be three concurrent focuses/aims of the manuscript 1) the off-host part of the life-cycle, 2) the role of the environment in interaction with host-parasite dynamics generally, including in-host (e.g. L456 and Box 6), and 3) the three different approaches that can be integrated to better address questions in future research and advance our understanding of

parasitism. Only 1) is stated explicitly by the authors as the focus of the review on L35 and L55, however it seems like more attention is paid to 2) the role of environmental factors more generally and in-host. Focus 3) is clearly mentioned more as a lens for considering the literature but I think is really part of the intended message to integrate disciplines and direct future research. I think all three of these aims can be, and are, achieved with the manuscript. I just feel like what I expected after reading the abstract and the first paragraph of the introduction, was not exactly the same direction the manuscript took later on, particularly in the perspectives section.

Response: This is such a great comment, clearly laying out the conflict between the stated aim and what the paper achieves, and then how to resolve it. Reining in the scope of this review was a challenge, given the gap between our questions of interest (points 1 and 2 noted by the reviewer) and the segmented approaches to how these studies have been conducted in different taxa (point 3). Relatively few studies explicitly tested the concepts that motivated us to write this review, while many, many ask questions relevant to those concepts, that warranted consideration. Doing justice to such a rich literature, while also highlighting its shortcomings for the questions raised here, has been a daunting task. To better represent the content of the paper, we added text to the abstract and introduction to note these several goals. (Our changes are detailed below in response to the next comment.)

I think the way forward is relatively simple and not much is required to address this. I think an addition or two to the introduction and/or some changes of phrasing and emphasis would achieve this clarity of focus, rather than any changes of content:

Aims 2), and especially 3) are not limited to the off-host part of the life-cycle, so it may be better to explicitly state that these are additional aims that would benefit studies of host-parasite dynamics more broadly – perhaps even in the abstract to set the reader’s expectation.

Response: We added additional text to the first paragraph of the introduction highlighting the multiple aims of this review (L55-63):

“This review therefore has two main goals. The first is to draw attention to the importance of the off-host portions of parasite life cycles in causing variation in host-parasite relationships and disease outbreaks. To achieve this, we broadly review how environmental context alters host-parasite dynamics, and hone in on under-studied parasite off-host survival traits and their ecological and evolutionary implications for virulence-transmission relationships. The second goal is to suggest improvements for how we investigate virulence-transmission relationships, by taking the environmental context of parasite transmission into consideration and including a greater diversity of host-parasite taxa.

We updated the abstract to reflect these changes as well (L34-37, 42-44). More on this second goal is added later in the introduction in response to reviewer 2 comments below.

A great deal of literature is explored to demonstrate the importance of environmental effects on host-parasite dynamics generally, and not just specifically during off-host stages. I initially thought the authors were perhaps conflating these general environmental effects with off-host process. The authors do later clearly acknowledge this distinction L231-237, but I feel that this acknowledgment should come sooner and be stated as part of the introduction for clarity, perhaps along with an explanation why environmental effects are explored at length, including in-host.

As I interpret it, the authors do this because they are exploring broader literature that addresses environmental effects at any life stage (through the 3 approaches) to highlight how the environment might plausibly affect specifically the off-host life stages – which has been understudied in the context of virulence and transmission (table 1). This set-up broadly speaking – to consider the environmental effects on off-host stages as an extended case of environmental effects generally – is also then a good primer for considering the off-host part of the life cycle as the long-arm of the same temporal spectrum between direct transmission and environmental transmission, where the environment affects host-parasite dynamics across all life stages, either in- or off-host (L174-177).

Response: This comment was helpful in framing our response to the reviewer's previous comment, in how we presented the goals of the manuscript. We kept the text mentioned in Approach II, since it follows from the literature presented in the first two approaches. Reviewer 2 also had a similar comment on the environmental context in general versus parasite persistence specifically.

I hope that I have articulated my thoughts clearly. My line-specific comments were (more or less) written at first read, but I hope they indicate the points at which the focus was unclear to me and could be differently emphasised.

Response: Your thoughts have been articulate and extremely helpful. We are grateful!

Line-specific Comments

Abstract

L43 This last sentence is slightly disconnected to the focus of off-host stages? This is a much broader application beyond off-host.

Response: The abstract was revised based on comments of both reviewers regarding the scope of the review. We think this now connects better.

Introduction:

L55 Could possibly explicitly state here something like “While the environment can influence host-parasite dynamics at any stage, here we extend this focus...”

Response: This line in the introduction was edited in response to the general comments above. We also adjusted the abstract accordingly.

L61-63 Virulence as a composite trait of both the parasite and the host?

Response: We clarified this as the “parasite-induced change in host mortality or morbidity” (L67-68), but left the details on why this isn't quite right for Box 2.

L81-89 This paragraph started broadly but limited environmental effects to climate and did not explore further biotic factors or non-climatic abiotic factors off-host. I think each existing sentence is appropriate here, but could be expanded upon slightly for a complete entry into

thinking about the off-host environment. Perhaps an example of a biotic interaction outside the host that would affect virulence and/or transmission?

Response: We added a sentence giving examples of biotic interactions to this paragraph (L93-95):

“Biotic factors also affect host-parasite interactions, including host variation and a variety of species interactions occurring inside and outside of hosts (e.g., [41-45]).”

L84-86 [separate point] These environmental factors are not necessarily affecting off-host life stages though? For example, ref [40], using climate data to predict Dengue outbreaks, but Dengue doesn't have an off-host stage? I think the climate variables here better predict availability of free-standing water required for mosquito reproduction, and consequently Dengue vertical transmission and proliferation in mosquitoes. If the focus of this review as stated on L55-57 is the off-host portion of the parasite life-cycle, then would it be better to use examples from parasites with a free-living, or motile life stages, or environmentally persistent propagules, if used in this context? Otherwise, is this conflating environmental factors that affect all life-stages with environmental factors that affect off-host stages? [This comment was written before reading L231-237; perhaps somewhere at the start of this paragraph or at L55 it would work to explain that you explore environmental effects on host-parasite systems generally, both in- and off-host in this review to understand the diversity of environmental impacts that could theoretically impact the off-host stages]

Response: At this point in the manuscript we are focused more broadly on parasites in general. We believe the changes made to the scope of the introduction in response to earlier comments have resolved the issue raised here.

Approach 1

L145 Rafaluk and Jansen?

Response: This was corrected.

L166-170 Is this potentially a typo with the relationship the wrong way around on L170? “survival in the environment is positively correlated with multiplication rate in host cells” and in the following sentence “the highest replication rates in cells (i.e. largest burst size) had the least persistence”; these seem to directly contradict one another? Is there some technical difference between multiplication and replication rates and an understanding of the burst size is important to this point?

Response: Great catch! This was a typo, and this was corrected (L186-187).

L182 This should be ‘effect’ not ‘affect’.

Response: Corrected.

Approach 2

L185 Should the comma in this section title be a colon or semi-colon. It doesn't quite read right with a comma.

Response: This was changed to a colon.

L189-191 For me, this sentence seemed a little counter intuitive as gene flow usually results in genetic homogeneity among populations. Having now read ref [71] to better understand, it could be good to add something explanatory to this sentence i.e. that gene flow actually increases genetic diversity available for selection which increases the efficiency of local adaptation.

Response: We completely agree. This finding also confused us initially. We added a sentence to explain this (L212-214):

“In considering host-parasite interactions, parasite local adaptation tends to occur if parasite gene flow is higher than host gene flow among populations [78, 79]. While this may seem counterintuitive, gene flow increases the genetic diversity available for selection, increasing the efficiency of parasite local adaptation.”

L222-223 Again, the off-host environment is stated as the main focus of the manuscript but is only mentioned here briefly in discussion of the larger environmental context that shapes host-parasite dynamics. I think what is written here is fine, but the emphasis earlier in the manuscript should change. L193

Response: The scope of the introduction was revised based on comments above.

L231-237 This is the important acknowledgement that environmental effects on the whole host-parasite system and environmental off-host effects are not the same thing. Perhaps this critical point/framework should come much earlier in the introduction as it helps avoid possible conflation of the environmental effect as a whole and in off-host stages. Doing so would also help to explain why you explore some in-host environmental effects at length, while the focus of the paper is really stated as the off-host environmental effects. This paragraph could then remain, but in a pared down form that simply describes the benefits of combining approaches 1 and 2.

Response: The scope of the introduction was revised based on comments above, but that did not include the text here on how to combine approaches 1 and 2. This particular paragraph seems better suited to be presented after we've reviewed the positives/negatives for both approaches.

Approach 3
No comments

Perspectives

L305-309 Perhaps here you should state explicitly that anthrax is a good example because it is an ETP with a highly persistent off-host life-cycle stage to keep with the off-host focus. [You do this in the Box]

Response: This was added (L347-349):

“We provide an example employing several aspects of this framework using anthrax in wildlife systems as a case study (Box 5), since *B. anthracis* is an ETP commonly invoked for the evolution of high virulence and high environmental persistence.”

L340-343 These two sentences seem contradictory. The first sentence here seems to infer missing terminology, but the second sentence seems to describe terminology that is commonly used for apparently this purpose. Why is this terminology inadequate for the purpose described, or is it being suggested here that this terminology should be adopted consistently and with the definition that e.g. an ETP that persists specifically in water before transmission to a host should be consistently be called a water-borne parasite, and ‘water-borne’ should be a term reserved only for these types of parasites?

Response: Our response to this is distributed in several places throughout the manuscript. First, we clarified these sentences to specify that transmission tends to be described by the specific reservoir, not a more general mode (L379-382):

“Yet, we lack a general terminology to describe transmission from an environmental reservoir to a host. Instead, this type of transmission tends to be described by the specific reservoir (e.g., water-borne, soil-borne, food-borne).”

We do, of course, call this “environmental transmission” throughout the manuscript. To clarify that this is not necessarily common terminology, we added “indirect transmission” to the glossary and included a description of environmental vs indirect transmission to Approach 1 where modes are first discussed (L143-145):

“Throughout this review we use the term environmental transmission to describe host infection from an environmental reservoir (others would call this indirect transmission, but this term has a broader definition, e.g. [60]).”

Then, under “methodological developments” we edited this sentence to clarify the issue with how transmission is binned among modes (L424-427):

“We need to reconsider how we prioritize the importance of parasite interactions with macroscopic organisms over microscopic organisms, and how we group parasites for evolutionary analysis based a transmission mode (e.g., environmental, direct, arthropod vector-borne) given that often alternative pathways are possible.”

L355 ‘between-host transmission processes...’ Would it be worth being explicit here that this includes everything on the spectrum from direct transmission to extended periods off-host?

Response: We added a sentence clarifying this (L396-398):

“This is true not just for ETPs, but across the spectrum of direct to environmental transmission as well as for other transmission modes.”

L395-398 This sentence was hard to digest. Could the same point be simplified/split into two sentences?

Response: This was edited to the following (L434-437):

“Thus, we may be biased by instinctively considering macroscopic secondary hosts, vectors and interactions in a different evolutionary light than non-mammalian “secondary hosts” and non-

arthropod “vectors” where the mutualists/antagonists are chiefly other microorganisms.”

L391-392 Is this what was meant by biotic environmental factors in the introduction at L83-84? Could this be mentioned briefly in the introduction?

Response: This comment was addressed in response to an earlier comment.

Boxes

L1051 It could be worth qualifying this and stating something like ‘Examples include many respiratory infections spread through coughing and sneezing etc., as well as...’. I didn’t immediately make the link between shedding and respiratory infections collectively.

Response: This clarification was added.

L1217 Could VNTR be lengthened to variable-number tandem repeat (?) as this is its first and only usage.

Response: Corrected.

Table 1: In the column headings would it not make more sense to have “Parasite” rather than “Pathogen” given the broad sense interpretation of Parasite throughout the paper?

Response: Good point. This was corrected.

Reviewer: 2

Comments to the Author(s)

GENERAL COMMENTS:

This paper is a review of research on transmission-virulence relationships, presented through the perspective that considering the effect of environmental persistence (the off-host stage) of the parasite can clarify existing inconsistencies and lead to novel and robust insights. The authors frame the existing literature based on three research approaches: studies of virulence-transmission trade-offs, studies of local adaptation, and studies using phylodynamics, and then discuss how these and other approaches can be integrated to yield new insights into disease dynamics and transmission-virulence relationships.

The first half of the paper does a nice job reviewing virulence-transmission literature within the framework of the three common approaches. The organization of research based on the three approaches is useful. The writing is clear and easy to follow. The thesis that environmental persistence must be considered in virulence-transmission trade-offs is important and worthwhile. These strengths argue for the value of this manuscript. However, while the thesis is clearly stated in the introduction and abstract, some sections blurred that point or seemed to conflate it with other arguments. For example, the importance of considering the effect of the environment on parasite traits is sometimes blurred with the importance of studying environmentally-transmitted parasites. Similarly, virulence-transmission and virulence-persistence seem to be used

synonymously in a section that discusses how persistence affects virulence-transmission relationships. (Therefore the use of terms is circular). See details below.

Response: Noted. We believe this comment has been addressed through the suggestions of Reviewer 1. Adding additional aims to the intro/abstract helped clarify the connection between the environmental context on the host-parasite interaction and specifically on survival of ETPs.

The authors propose integration of the three approaches to achieve clearer and more consistent results regarding transmission-virulence trade-offs (Figure 1). For example, findings from Approach II (effects of environment on parasite traits or host-parasite dynamics) should be integrated into Approach I for a more robust and consistent understanding of virulence-transmission relationships. This is a useful framework overall. However, the final step in this framework (D in Figure 1) is a bit vague. Admittedly, this vagueness could stem from the fact that this final step depends on the specifics of the study system. However, in that case the authors can, and I argue should, provide a concrete example of this last step in their case study of Anthrax (Box 5). As is, the case study provides only a general statement that the results of the previous steps in the study would be used to “build models”.

Response: We agree that concrete examples here would be valuable, and that the outcomes may be system specific. This is something we are working toward in our own research, but do not yet have answers. In an effort to clarify this component, we reworded the legend of Figure 1: “D) Finally, once genotype-phenotype relationships, and how these vary with environmental variables, are described for the disease system, use that understanding to build statistical and theoretical models of virulence-transmission relationships. These models can test for trade-offs in parasite life history traits, and determine if there are common environmental factors shaping the outcome of these relationships across space.”

We also expanded the text at the end of Box 5 (L1296-1300):

“Then, we can test mathematical models of transmission and interactions between life history traits that may lead to the outcomes observed in different disease systems. Ideally, for anthrax or other host-parasite systems, inferring common drivers of the variation in host-parasite relationships across environmental gradients will allow us to infer properties of these systems that scale beyond the specifics of any particular environment, host species, or parasite species.”

The second half of the paper (“Some perspectives on transmission and virulence in natural populations”) is informative. However, it is a bit of a departure from the first half of the paper and the thesis of considering environmentally transmitted parasites (ETPs). It covers a very broad spectrum of recommendations, including the need for interdisciplinary research, new conceptual developments, and new methods (e.g. metagenomics, studies at large spatial scales, improvements to niche models, and genome-wide association studies.) Again, this is useful, it just seems extremely broad and somewhat disconnected from the thesis and structure of the first half of the paper.

Response: To better represent the full scope of the paper, we added the following text to the introduction (L115-122):

“In the second half of this review, we offer some perspectives on how to integrate these approaches to investigate virulence-transmission relationships across the range of biodiversity in host-parasite systems. This section focuses broadly on the pivotal role of the environmental context in disease processes, and specifically on parasite dynamics in the off-host environment. We note the continued need for more interdisciplinary research, as well as conceptual and methodological developments in combination with advances in life history evolution, phylogenetics, adaptive dynamics, and comparative genomics, to improve our understanding of virulence-transmission relationships.”

In addition, we added a paragraph to the start of the second half of the manuscript, to help bring these two halves of the manuscript together. This is partly new text, partly ideas pulled out of the previous Box 3 (L317-331):

“In the previous sections we reviewed three research approaches used to study the evolution of virulence and transmission, highlighting how the environment can have important effects on virulence-transmission relationships. Within this environmental context, we focused specifically on parasite survival during the off-host life stage. Perhaps counterintuitively, an off-host perspective allows us to conceptually merge direct and environmental transmission. These two transmission types are not separate processes but occur along a time continuum in the off-host environment [76], and should ideally be described by the same unified theory. This is especially true given that many classic examples of directly transmitted parasites are found to have the capability of persisting in the environment for longer than thought (e.g., the *Mycobacterium tuberculosis* complex [118]), have diversity in environmental niches among strains that can affect their infectiousness or virulence (avian influenza [119]), or may show the potential for an environmental reservoir long thought not possible (e.g., *Yersinia pestis* [120, 121]). In addition, even directly transmitted respiratory infections, considered short lived in the environment, have fascinating mechanisms to increase their survival and dispersal in the off-host stage [122, 123]. “

Overall, I found this to be an interesting and informative review. The argument that we need to consider ETPs, and to view them on a continuum rather than as fundamentally different from directly transmitted parasites, is broadly important to the study of disease dynamics and evolution and is well substantiated within the manuscript. My critiques are mostly aimed at tightening and clarifying the manuscript. I might also suggest breaking the paper into two articles based on the weaker connection between the first and second half - however, this issue is partly a matter of preference regarding the organization of review articles, and should be left up to the editor and authors to assess.

Response: We appreciate both reviewers noting this gap. We believe the additions to the introduction and the perspective sections have clarified the full scope of the manuscript. This better represents its content, and helps hold together the two parts of the manuscript, and has helped hone our message.

SPECIFIC COMMENTS:

1. Figure 1. What is a “virulence-transmission” trait? The conceptual figure shows the example of the shape of the curve of N vs survival. However, it’s hard to imagine how this is a directly measurable trait. I also can’t think of any other “virulence-transmission” traits. Do you mean

virulence traits and transmission traits, as separate entities? Please clarify.

Response: Yes, this should be “virulence and transmission traits” and has been edited in the figure.

2. Section “Approach I” argues that studies examining virulence-transmission trade-offs often fail to consider the effect of environmental variation on the virulence-transmission relationship. By controlling transmission, these studies miss the effect of environment. This section was generally well written and informative, but two points need clarification:

2a. It was unclear how studies “control transmission”. Please specify what you mean.

Response: We added the word “standardize” to clarify this:
“Most empirical tests of theory using ETPs control for the external environment and standardize the transmission phase, to focus on coevolutionary dynamics between host and parasite.”

2b. The section seems to conflate the concept of a transmission-virulence relationship with a persistence-virulence relationship (e.g. Line 110 vs Line 152). This is problematic because, as I understand it, a central point of the manuscript is that environmental persistence AFFECTS the transmission-virulence relationship, so transmission and persistence cannot be synonymous.

Response: In the third paragraph of this section when we move from general expectations for virulence-transmission relationships to those specifically for ETPs, we added a sentence to make this clear:

“For these parasites, the transmission component most often considered is parasite survival, as a distinctive trait of these parasites and one that is relatively easy to quantify in experimental studies.”

We also discuss this in the newly focused Box 3. Persistence is one aspect of transmission that is more easily measured, akin to parasite growth rates as an aspect of virulence.

3. Section “Approach II” reviews studies of local adaptation or geographic variation in parasite traits or host-parasite dynamics. The authors conclude that such studies of spatial variation should be used to inform transmission-virulence studies (Approach I). Approach I would be strengthened by considering how environmental variation affects parasite traits and transmission. This section is good and I have only one comment here:

3a. This section switches between discussion of two topics (1) environmental effects on traits and (2) environmental persistence. The distinction between these two is at times blurred and confusing.

Response: Reviewer 1 similarly commented on this tension in the manuscript between the effect of the environment on the host-parasite relationship more generally, and environmental persistence specifically. To address these comments we added text to the abstract, introduction and perspective sections to clarify this, with how we addressed these in detail listed under Reviewer 1’s comments above.

3b. L. 231-235. This statement suggests that the whole section (Approach II) is about host-parasite co-evolution (geographic mosaic theory), not parasite traits. I disagree: some of the statements in this section discuss traits such as virulence, growth, or infectivity.

Response: This is a good point. We clarified this as follows:

“It is important to remember that many of the studies detailed under Approach II examine host-parasite interactions in the context of environment variation, and not specifically how environmental variation i) affects parasite survival traits in the off-host environment and ii) how those survival traits are related to infection or virulence traits.”

4. Section “Approach III” reviews phylodynamic approaches to inferring disease dynamics, population size, or transmission. Studies of functional genes can detect evolution of virulence determinants. The authors emphasize that interpretation of phylodynamic studies becomes uncertain with environmentally transmitted parasites (ETPs).

5. BOXES:

The “point” of Box 3 is not clear. It starts by pointing out that R_0 is used oversimplistically and that it’s important to consider time. It goes on to note that virulence and transmission are conflated in many studies. It ends by emphasizing the importance (and current neglect) of environmental persistence and the need for a unified theory that sees environmental persistence along a continuum. These are all good points, but the point of the box overall is unclear.

Response: Box 3 was restructured, to focus on the challenges of defining and measuring transmission, and several problematic simplifying assumptions that are used as a result of these challenges. Doing so caused us to break up/shorten Box 2, which is now focused solely on virulence, while Box 3 focuses on transmission.

In general, all of the boxes are too long, each containing a full page or more of single-spaced text. Boxes should be concise presentations of concepts or issues relevant to the main body of the manuscript. I recommend condensing the content of each box to clearly and efficiently convey their points.

Response: The boxes add depth to important points in this review that would otherwise take away from the flow of the main text, and we do think these are important to include. However, we edited and reduced text length for all of the boxes. Currently the longest one is the glossary.

6. Table 1: This is an informative Table. However, what is meant by “alternate strategies evolved” under “Virulence-transmission relationship”?

Response: This should have stated “alternative strategies evolved.” To simplify this column, we replaced “alternative strategies evolved” (from experimental studies) and “variable outcomes” (from theoretical studies) to “context-dependent outcomes.” The separate designations were really a function of the study type, which is already included in the table.

7. L. 392. change “may be vectored by other microbes” to “may be vectored by other organisms”? The first example is of copepods vectoring *V. cholerae*.

Response: Microbes was changed to microorganisms.

8. L. 393 *Vibrio cholerae* is misspelled as “*Vibrio cholera*”)

Response: Corrected.

9. L. 1217 Write out Variable Number Tandem Repeats at the first use of the acronym.

Response: Corrected.